# Microbial co-occurrences on catheters from long-term catheterized patients

Taylor M. Nye [1,5], Zongsen Zou[1,5], Chloe L. P. Obernuefemann[1], Jerome S. Pinkner[1], Erin Lowry[1], Kent Kleinschmidt[1], Karla Bergeron[2], Aleksandra Klim[2], Karen W. Dodson[1], Ana L. Flores-Mireles [3], Jennifer N. Walker [4], Daniel Garrett Wong[2], Alana Desai[2], Michael G. Caparon[1] ✉ & Scott J. Hultgren [1] ✉

Catheter-associated urinary tract infections (CAUTIs), a common cause of healthcare-associated infections, are caused by a diverse array of pathogens that are increasingly becoming antibiotic resistant. We analyze the microbial occurrences in catheter and urine samples from 55 human long-term catheterized patients collected over one year. Although most of these patients were prescribed antibiotics over several collection periods, their catheter samples remain colonized by one or more bacterial species. Examination of a total of 366 catheter and urine samples identify 13 positive and 13 negative genus co-occurrences over 12 collection periods, representing associations that occur more or less frequently than expected by chance. We find that for many patients, the microbial species composition between collection periods is similar. In a subset of patients, we find that the most frequently sampled bacteria, *Escherichia coli* and *Enterococcus faecalis*, co-localize on catheter samples. Further, co-culture of paired isolates recovered from the same patients reveals that *E. coli* significantly augments *E. faecalis* growth in an artificial urine medium, where *E. faecalis* monoculture grows poorly. These findings suggest novel strategies to collapse polymicrobial CAUTI in long-term catheterized patients by targeting mechanisms that promote positive co-associations.

Healthcare-associated infections (HAIs) represent a significant public health burden, where an estimated 1 in 31 patients that enter a hospital will go on to develop an HAI, resulting in an additional $28.4 billion in healthcare-related expenses and 100,000 deaths[1,2]. Among HAIs, urinary tract infections (UTIs) are the most common, affecting over 400 million people worldwide in 2019 alone[3]. UTIs can be distinguished as complicated or uncomplicated. Up to 75–85% of uncomplicated UTIs are caused by uropathogenic *Escherichia coli*

(UPEC)[4–6]. Unlike uncomplicated UTIs, catheter-associated UTI (CAUTI) is caused by a diverse range of pathogens, including UPEC (23.9%), fungal *Candida* spp. (17.8%), *Enterococcus* spp. (13.8%), *P. aeruginosa* (10.3%), and *Klebsiella* sp. (10.1%)[7,8]. Uncomplicated UTIs (uUTIs) predominantly affect pre-menopausal and non-pregnant women and occur in otherwise healthy individuals with no functional or structural abnormalities of the kidneys or urinary tract[4]. Complicated UTIs (cUTIs) occur in individuals with additional risk factors,

[1]Department of Molecular Microbiology and Center for Women's Infectious Disease Research, Washington University School of Medicine, Saint Louis, MO 63110-1093, USA. [2]Department of Surgery, Division of Urologic Surgery, Washington University School of Medicine, Saint Louis, MO 63110, USA. [3]Department of Biological Sciences, University of Notre Dame, Notre Dame, IN 46556, USA. [4]Department of Microbiology and Molecular Genetics, McGovern Medical School, The University of Texas Health Science Center, Houston, TX 77030, USA. [5]These authors contributed equally: Taylor M. Nye, Zongsen Zou. ✉e-mail: caparon@wustl.edu; hultgren@wustl.edu

including underlying health conditions and abnormalities or obstructions in the kidneys and urinary tract, pregnancy, and intermittent or long-term catheterization[4].

It is estimated that 15·25% of hospitalized patients will receive a urinary catheter and that 75% of healthcare-acquired UTIs are associated with catheterization[9]. Furthermore, the risk of bacterial colonization of a catheter increases 3–7% per day upon placement, with the risk of catheter colonization and associated complications near 100% in long-term catheterized patients[10,11]. The treatment of UTIs has been complicated by the rise of antimicrobial-resistant (AMR) uropathogens, many of which were cited as urgent or serious threats in the CDC's 2019 Antibiotic Threats Report. These AMR pathogens were classified as: (i) urgent, carbapenem-resistant *Acinetobacter* and carbapenem-resistant Enterobacterales; and (ii) serious, drug-resistant *Candida*, ESBL- Enterobacterales, vancomycin-resistant enterococci (VRE), multidrug-resistant *Pseudomonas aeruginosa*, and methicillin-resistant *Staphylococcus aureus* (MRSA)[12]. Indeed, a 2019 report listed UTIs as one of the leading global causes of AMR-associated deaths[13]. As these uropathogens become increasingly antibiotic-resistant, new treatment strategies are necessary to combat the rise of antibiotic-resistant UTIs.

Another factor complicating the treatment of CAUTIs is the high prevalence of polymicrobial catheter colonization, with 31–87% of catheters and urine from catheterized patients colonized by two or more species, depending on the study[14–16]. Polymicrobial bacteraemic UTIs are associated with increased mortality relative to bacteraemic UTIs caused by a single uropathogen[17]. The relationships between the bacteria in these polymicrobial communities is still poorly understood, including the mechanistic drivers of the positive species interactions that promote species co-occurrence and the negative species interactions that prevent co-occurrence in the same microenvironment. During urinary catheterization, the host wounding response results in the deposition of host proteins, including fibrinogen, onto the catheter that can be used by several bacterial species as a food source and as a means to adhere to and form biofilms on the catheter[18–20]. Positive co-occurrences between species may thus result from pioneer species that are able to colonize the catheter and then recruit other microbial species, forming polymicrobial communities. While this has been demonstrated to occur in the polymicrobial communities of oral biofilms, wherein certain groups of bacteria, including streptococci and Actinomyces species, serve as the pioneering species that can adhere to enamel pellicle and facilitate the colonization of other microbial species, the establishment of pioneer species and the subsequent effects on colonization in CAUTI has yet to be fully elucidated[21,22]. Within these communities, "cross-feeding" or "cross-signaling" may occur to augment the growth and/or biofilm formation of one or more of the species. Further, it has been shown that *E. faecalis* can suppress the host innate immune response, allowing for augmented growth of *E. coli* in a co-infection model of CAUTI[23]. Negative co-occurrences may be due to competition for nutrients or alterations in host response. For instance, MRSA elicits a robust host response which may result in decreased co-colonization by other bacterial species that are unable to withstand this response[24]. Understanding the mechanisms that drive the formation and persistence of these communities is essential to designing new therapeutics that target the bacterial interactions that promote CAUTI.

Here we analyze the polymicrobial community composition on catheters of long-term catheterized patients. We collected a total of 366 catheter and urine samples from 55 patients over an average of 6–7 -monthly collection periods. We discovered that ~80% of the collected samples are polymicrobial, however most of these samples only contained 2–3 detected bacterial species. We performed co-occurrence analysis to determine the genera that co-occur in samples more and less often than expected by chance, identifying several positive and negative significant pairwise genus interactions among common uropathogens. We also observed clustering of the microbial species composition between sample periods within the same patients in a principal coordinate analysis, suggesting similarities in the community composition upon several re-catheterization events within the same individual. For multiple patients, we observed these long-term co-occurrences between the most common uropathogen, UPEC, with the most sampled Gram-positive opportunistic pathogen *E. faecalis*. We show that these species co-localize on patient catheters and that co-culture of clinical UPEC isolates dramatically increases the growth of clinical *E. faecalis* isolates in artificial urine medium. These results suggest that the urinary tract environment may be a significant contributor driving positive and negative interactions and must be considered in future in vitro analysis of polymicrobial CAUTI.

## Results

### Sample collection from long-term catheterized patients and demographics

We collected catheter and urine samples approximately monthly for up to 12 collection periods from enrolled non-hospitalized, long-term catheterized patients at the Barnes-Jewish Hospital System that were subsequently cultured for bacterial growth and identified via 16 S rRNA sequencing (Fig. 1a)[25]. We enrolled 55 patients, 20 females and 35 males, at an average age of 64 years at the time of enrollment (Fig. 1b). The reasons for catheterization included bladder outlet obstruction ($n = 18$), peripheral or central neuropathy ($n = 22$), rectovesical or rectourethral fistula ($n = 4$), stress urinary incontinence ($n = 4$), atonic bladder ($n = 6$), and temporary post-operative issues ($n = 1$, with catheter use exceeding 50 days). For each patient, we collected an average of ~7 samples, which consisted of catheter and urine samples at a given time point, with average catheter dwell times of $29.7 \pm 10.4$ days (Fig. 1c and Supplementary Fig. 9).

Inclusion criteria included patients that were scheduled to undergo removal of the indwelling urinary device(s) (such as catheters, both urethral and suprapubic, ureteral stents, etc.), being 18 years of age or older at the time of consent and being willing and able to provide informed consent. There were no exclusion criteria. As the long-term catheterized patients came into the clinic to have their catheters replaced, the removed catheter was placed into a sterile bag and urine from the visit was collected in a sterile cup (Fig. 1a, see "Methods"). The tip of the catheter (0.5 cm) and 10 µL of urine were cultured in Brain Heart Infusion (BHI) agar, as previously described[25]. For the catheter and urine samples, colonies were distinguished based on unique size, color, and morphology, resulting in 1–9 colonies selected per sample, and single colonies were cultured overnight and subsequently archived. Genomic DNA was then purified from each isolate for species identification via 16 S sequencing (see "Methods").

### Antibiotic use in long-term catheterized patients

Most patients (40/55, 72.7%) received at least one course of antibiotics over the course of the study, and nearly 50% (27/55) received more than one course of antibiotic and more than one type of antibiotic (Fig. 1d). The most prescribed antibiotics among our patient cohort included those commonly or specifically used to treat urinary tract infections, such as nitrofurantoin, methenamine, the fluoroquinolone antibiotic ciprofloxacin, the combination of trimethoprim and sulfamethoxazole, and the combination of the ß-lactam antibiotic amoxicillin and the ß-lactamase inhibitor clavulanic acid. The specific antibiotics prescribed over the course of catheterization for each patient are indicated in Supplementary Figs. 1–7.

### Identification of bacterial isolates in long-term catheterized patients

Overall, we detected 88 different species representing 32 genera of bacteria. Enterococcus was the most common genus detected, appearing in 192 samples, followed by Escherichia (104), Klebsiella

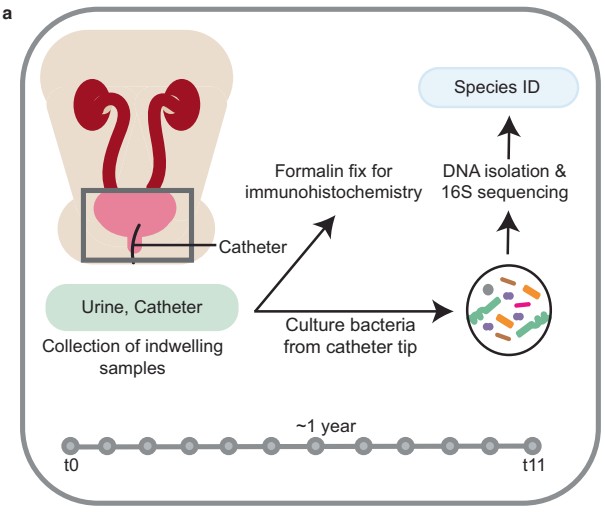

| Number of patients | 55 |
|---|---|
| Total sample collections | 366 |
| Average collections per patient | 6.7 ± 4.5 |
| Sex | |
| Female | 36.4% |
| Male | 63.6% |
| Average age | 63.4 ± 15.5 |
| Race | |
| Asian | 1.8% |
| Black | 29.1% |
| White | 67.3% |
| Other | 1.8% |

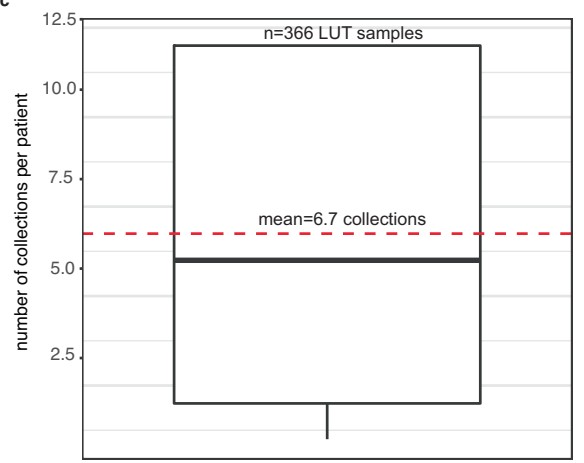

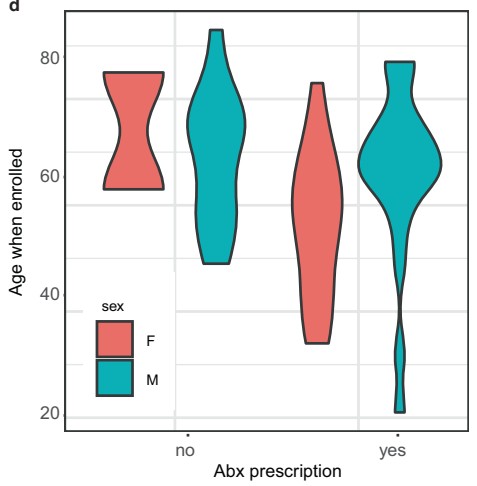

**Fig. 1 | Identifying the temporal microbial community composition from the urine and catheters of long-term catheterized patients. a** Study design. Catheter and urine samples were collected approximately monthly from long-term catheterized patients during catheter exchange. A portion of the catheter was fixed in formalin for immunohistochemistry. The microbes from the catheter and urine samples were cultured and identified via a 16 S sequencing approach. **b** Patient demographics. **c** Boxplot of average collections per patient included in the cohort. The bottom and top ends of the gray rectangle indicate the lower and upper quartile for the number of collections per patient, respectively, with the median indicated within the rectangle. The whiskers indicate the minimum number of collections per patient that are not outliers (within 1.5x the interquartile range). Average (mean) collection periods per patient is indicated by a dashed red line. **d** Violin plots representing patient age and antibiotic use over the course of the study.

(98), Proteus (95), Pseudomonas (93), and Staphylococcus (86) (Fig. 2a and Supplementary Fig. 8). To better understand the temporal aspects of genera detection in long-term catheterized patients, we examined the frequency (the proportion of the selected genera vs the total genera sampled for a specific collection period) of genus detection at the initial collection period ($t = 0$) through the final collection period ($t = 11$) for patients that had 9+ collection periods. We found that while Enterococcus species were sampled more frequently during the first couple of collection periods for an individual, the frequency of detection for Escherichia species remained fairly constant throughout the course of a particular individual's time in the study (Fig. 2b).

As the use of urine is the current clinical standard for the detection of bacteria in the urinary tract, we examined whether genus presence in the urine was concordant with genus detection on the catheter. We found that while species from Enterococcus and Escherichia were simultaneously detected on both the catheter and in urine, they were rarely detected in the urine alone. In contrast, species

from Pseudomonas were frequently detected in the urine alone (Fig. 2c). Further, we found that multiple genera were detected over the collection periods for most patients (Fig. 2d), which together prompted us to further examine the microbial species composition and dynamics of catheter and urine samples from our long-term catheterized patient cohort.

## Catheter and urine samples from long-term catheterized patients are polymicrobial

The most frequently identified genera were typically dominated by a single species. For example, the majority of enterococcal species were *E. faecalis* (187/192), Escherichia were all *E. coli* (104), and Klebsiella were predominately *K. pneumoniae* (80/98) (Fig. 3a and Supplementary Fig. 8). For each individual patient, we detected a median of six different species (5.95 mean) cumulatively over all collection periods, ranging from 1 to 16 species per patient (Fig. 3b and Supplementary Fig. 9). Given our data set consisted of several collections per each patient, which were representative of individual collection periods

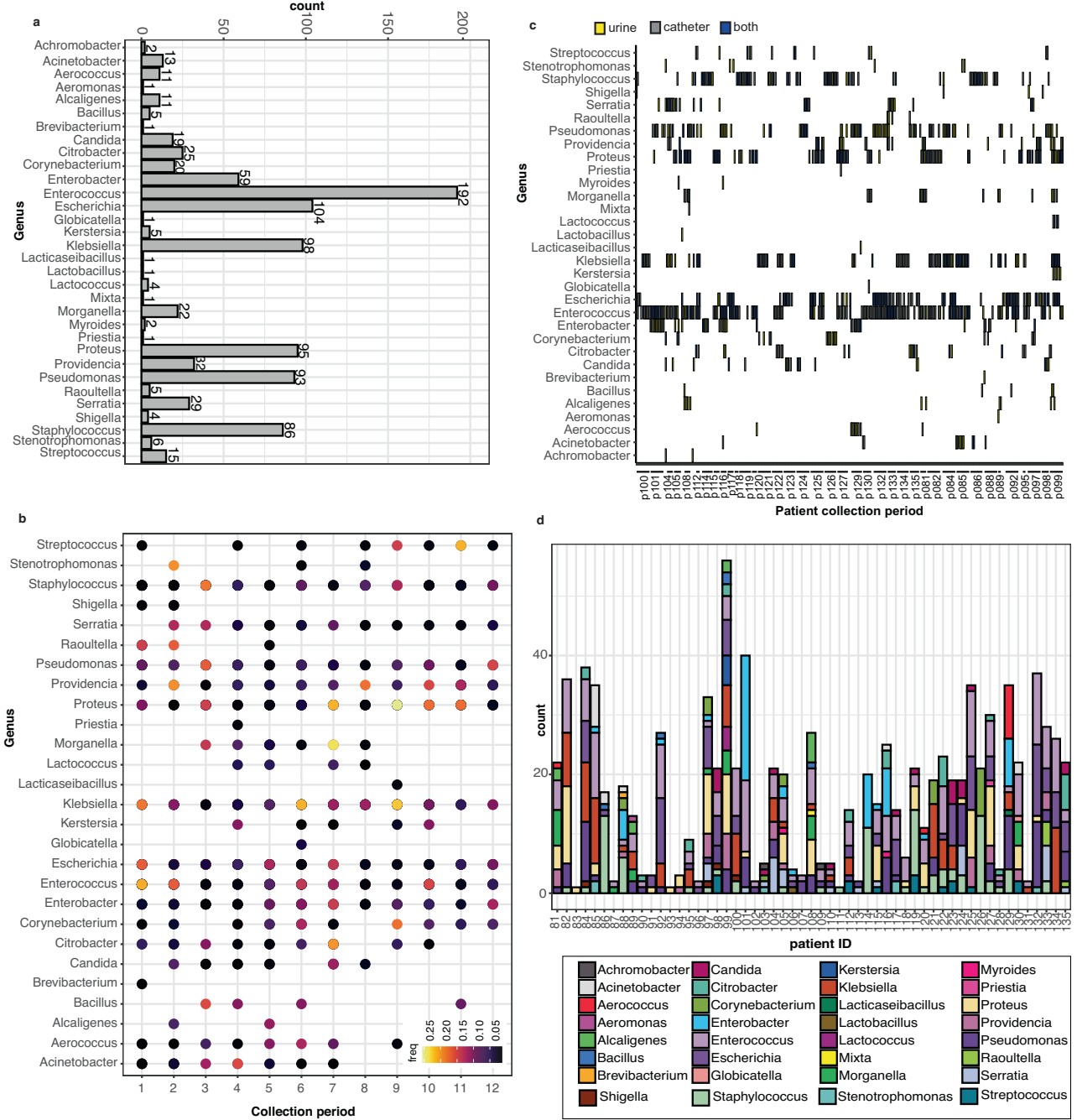

**Fig. 2 | Multiple genera detected per patient in long-term-catheterized patient cohort. a** Count data of the indicated genera detected in the catheter and urine samples in the long-term catheterization cohort. **b** The frequency of indicated genera detected over sample collection periods for patients with 9 or more collection periods. Higher frequencies are indicated in yellow while lower frequencies are in black. **c** The site of collected genera isolated from long-term catheterized patients. Each individual collection period is listed in order (0–11) for the indicated patients. The genera identified by 16 S sequencing is indicated on the y axis. Genera isolated from urine are colored yellow, from a catheter are colored in gray, and from both catheter and urine are colored in blue. If a genus is not present during a collection period, it is white. **d** Detected genera, indicated by colored key, by patient over all collection periods. Source data are provided as a Source Data file.

over consecutive time points, we sought to understand the dynamics of the microbial composition for individual samples within a single time point. For all collected samples, including urine and catheter samples from each time point, we detected two or more species in ~80.1% (296/366) of samples, concordant with current reports that most lower urinary tract samples are polymicrobial. There were 70 single-species samples collected across 28 different patients (Supplementary Table S1). Most samples with a single species detected contained species from the Staphylococcus genus (37.1%), with representatives from *S. epidermidis* (65.4%) and *S. aureus* (19.2%) in the

majority of the samples. Of the other single-species samples detected, 42.8% were common CAUTI uropathogens including *E. faecalis, E. coli, K. pneumoniae,* or *P. aeruginosa* (Supplementary Table S1).

While most of the samples were polymicrobial, we found that the median number of species detected for a given individual at a given collection was 2 (average 2.6), with a range from 1 to 8 species detected, contrasting the 6 species identified per patient over all collection periods (Fig. 3c). The first and third quartiles contained two and three species per catheter. Thus, most samples were polymicrobial, but had less than three species detected.

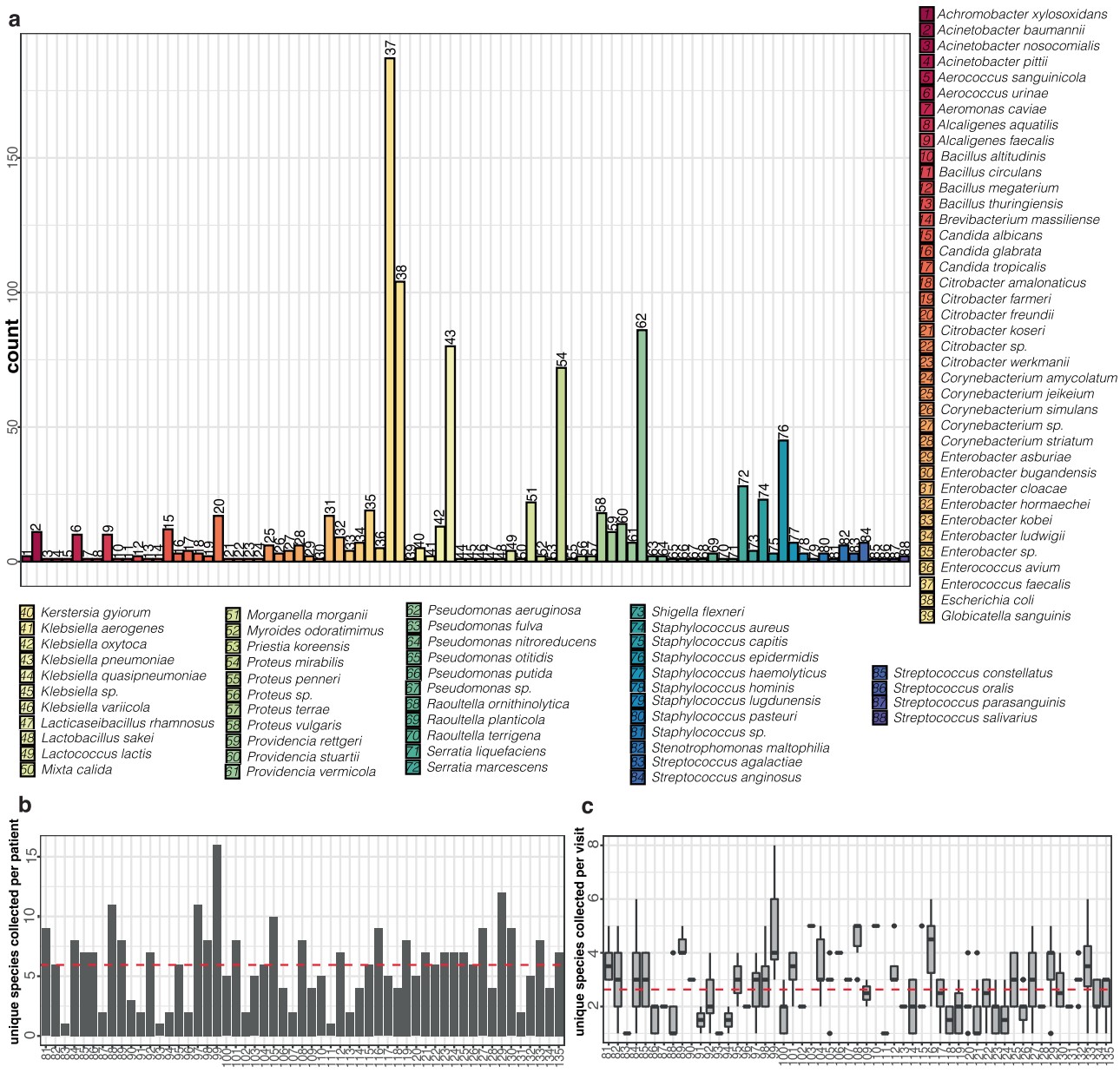

**Fig. 3 | The polymicrobial community composition of long-term catheterized patients averages 5–6 species per patient but only 2–3 species per collection period and are dominated by species that are frequently uropathogenic.**
**a** Number of occurrences of indicated species per patient per collection period detected in the catheter and/or urine samples in the long-term catheterization cohort. Numbers above bars correspond to numbered species key. **b** The number of unique species identified per patient. Average (mean) of species identified across all patients (*n* = 55) is indicated by a dashed red line. **c** Boxplots of the number of

unique species identified per patient per collection period. The bottom and top ends of the gray rectangle indicate the lower and upper quartile for the number of species identified per collection, respectively, with the median indicated within the rectangle. The whiskers indicate the maximum and minimum species detected that are not outliers (within 1.5× the interquartile range). For patients with 2 or fewer collection periods, the number of unique species identified are indicated by dots. Average (mean) of species identified across all collection periods (*n* = 366) is indicated by a dashed red line. Source data are provided as a Source Data file.

## Detection of positive and negative pairwise genera co-occurrence

We performed co-occurrence analyses to determine the pairwise genera that had positive and negative co-occurrences, defined as genera that were positively or negatively associated more frequently than would be expected by chance. The co-occurrence analyses were binned by collection periods (0–11), ensuring sample independence between patient collections over multiple time points[26]. For this analysis, genera were identified as "occurring" during a single collection period independent of the source of isolation (catheter, urine, or both). Our analysis identified 13 positive interactions and 13 negative

interactions (*P* value < 0.05) with an expected co-occurrence greater than 0 over all collection periods (Fig. 4a–l and Supplementary Tables S2–13). Of the significant negative interactions, where genera co-occur less often than would be expected by chance, the majority (11/13) involved Staphylococcal species repelling both Gram-positive (Enterococcus) and Gram-negative (Proteus, Pseudomonas, and Klebsiella) species. These data are consistent with the previous observation that the majority of samples that were not polymicrobial (e.g., only a single species detected) were from the Staphylococcus genus. The other negative interaction occurred between Proteus and Pseudomonas at collection periods zero and one.

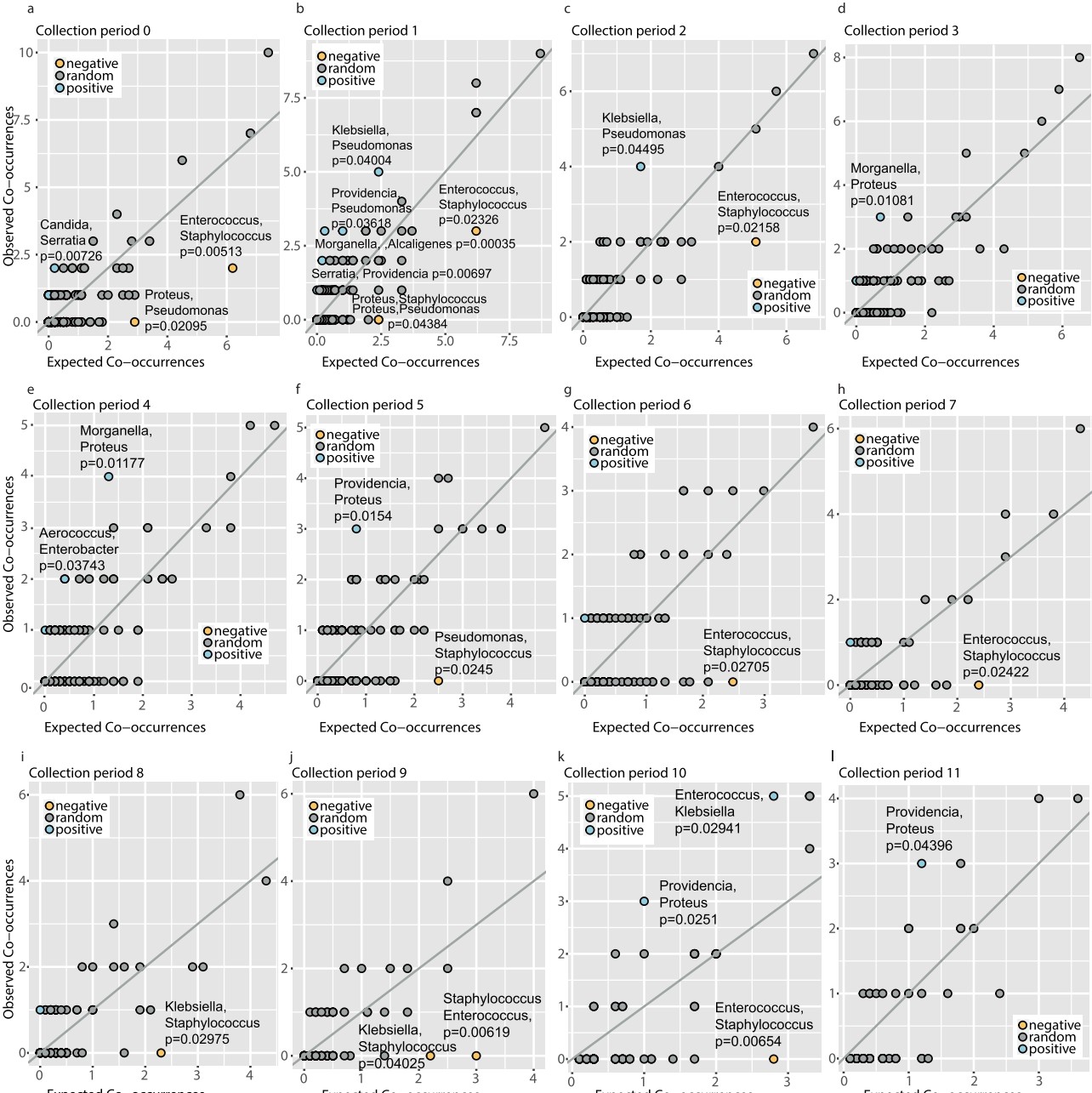

**Fig. 4 | Identification of positive and negative pairwise species co-occurrences from lower urinary tract samples of long-term catheterized patients.** **a**–**l** Number of expected co-occurrences (*x* axis) and observed co-occurrences between each pair of genera during the indicated collection period. Negative pairwise genera co-occurrences are indicated in gold, positive in blue, and random interactions are in gray. For each significant interaction with an expected co-occurrence greater than 0, the pairwise genera interaction and significance are indicated. The significance is based on the probability that two species co-occur at a frequency greater or less than the observed co-occurrence frequency, which can be used as *P* values[26]. Source data are provided as a Source Data file.

For the positive co-occurrences, with expected co-occurrence greater than 0, the most common were between the Proteus genus with Providencia (detected at collection periods 5, 10, and 11) and Morganella (detected at collection periods 3 and 4). We also detected positive interactions between the Klebsiella genus with Pseudomonas at early collection periods (periods 1 and 2) and Enterococcus at a late collection period (period 10). Additional positive co-occurrences included Candida and Serratia (collection period 0), Providencia with Pseudomonas and Serratia (collection period 1), Morganella with Alcaligenes (collection period 1), and Aerococcus with Enterobacter (collection period 4).

## PCoA reveals similarities of species occurrences within patients over successive collection periods

To determine the similarity of successive microbial community compositions isolated from the same patient over time we performed a Principal Coordinate Analysis (PCoA). We found that the major species driving the variance in community composition were *E. faecalis, E. coli, P. mirabilis, S. epidermidis* and *S. aureus* (Fig. 5a). We also observed clustering of collection points between the same patient, suggesting that the communities isolated from the same patient were very similar through multiple re-catheterization events (Fig. 5b). This was particularly evident for Patient 132, where all 12 collection periods clustered in

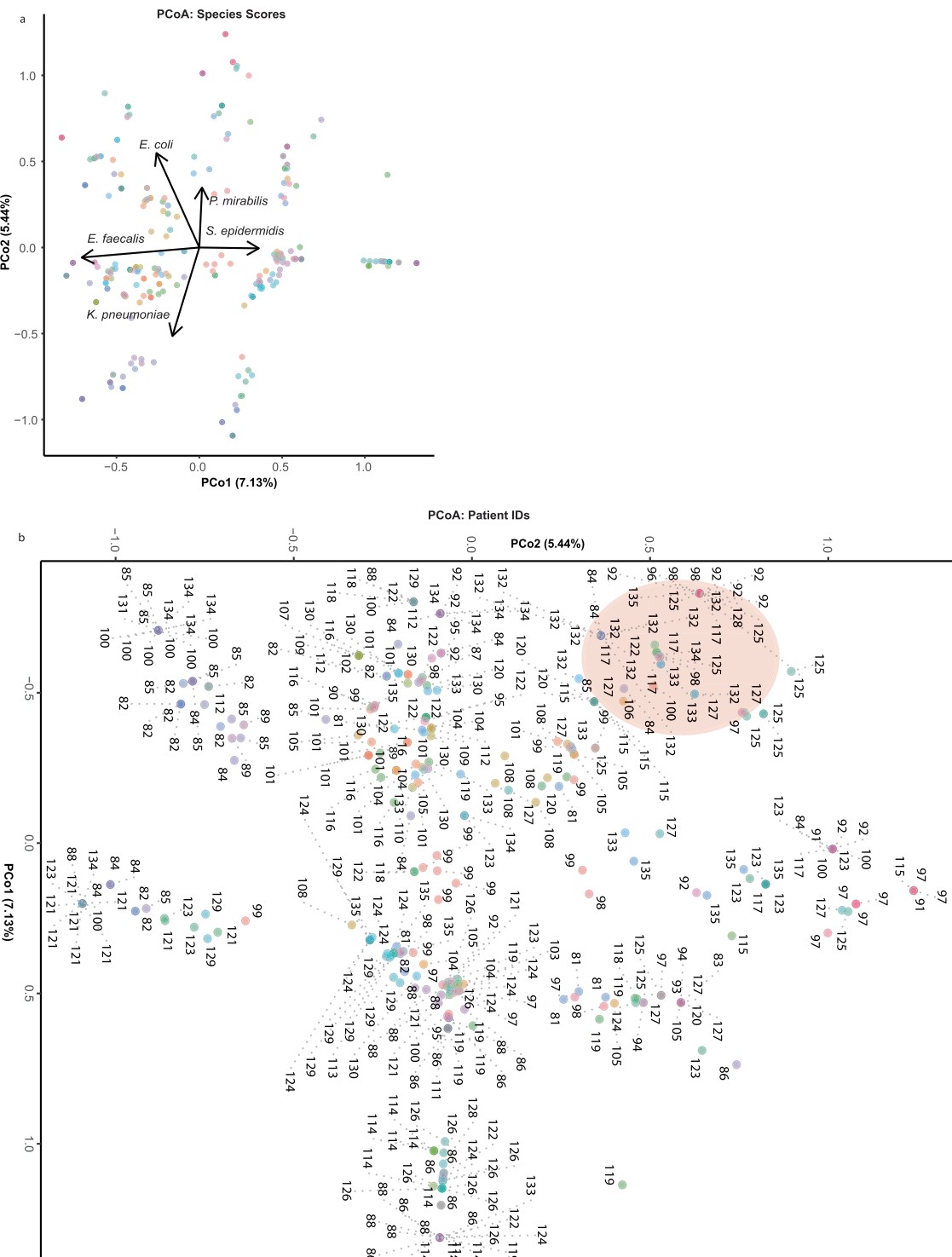

**Fig. 5 | Principal Coordinate Analysis of microbial community composition for each individual collection period.** PCoA of microbial community composition, based on presence/absence data of all species identified in the long-term catheterized patient cohort for each individual collection period. Points represent individual collection periods that are colored by patient ID. **a** Species scores are indicated by arrows in the PCoA. **b** Patient IDs are noted for each point, with dashed gray lines extending to patient ID number where points overlap. The cluster of points representing all 12 Patient 132 collection periods is shaded in red. Some individual collection points overlap on the plot where community composition was very similar/identical. Source data are provided as a Source Data file.

the same area of the PCoA, suggesting that the polymicrobial communities isolated from this patient were not drastically altered by subsequent re-catheterization events. Indeed, we found that Patient 132 was colonized by *E. faecalis* and *E. coli* over all 12 collection periods, suggesting a positive relationship between these species in the lower urinary tract environment.

## Co-culture of *E. coli* and *E. faecalis* stimulates *E. faecalis* growth in artificial urine media

The co-occurrence of *E. faecalis* and *E. coli* isolates at all collections in Patient 132 suggests that co-colonization may benefit one or both species in the lower urinary tract (Fig. 6a). To determine if the *E. faecalis* and *E. coli* strains were co-localizing on the catheter, we

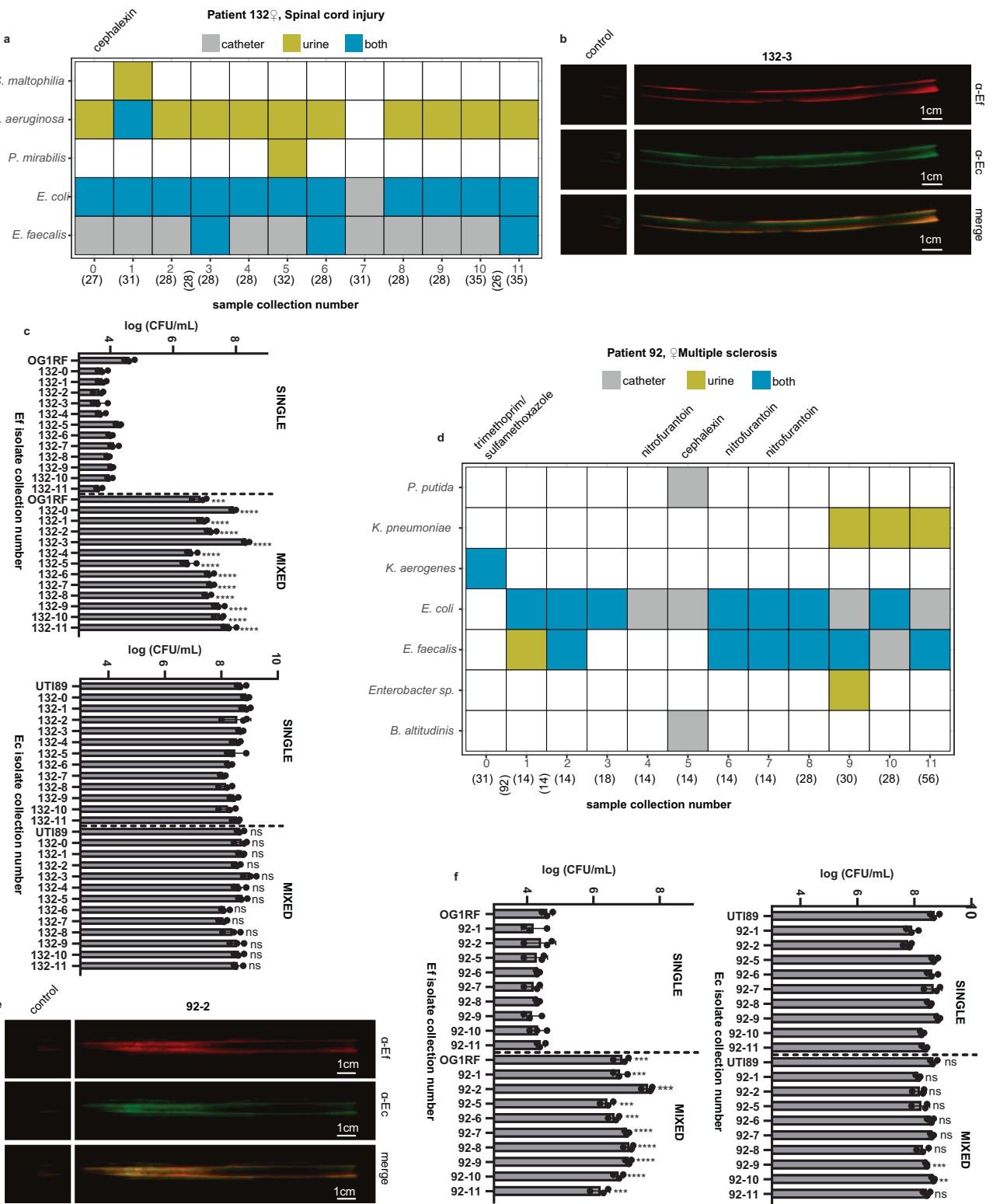

**Fig. 6 | *E. faecalis* and *E. coli* co-localize on patient catheters and *E. coli* clinical isolates augment growth of *E. faecalis* clinical isolates in AUM.** Clinical isolates collected from Patient 132 (**a**) and Patient 92 (**d**). Collection period is indicated on the *x* axis with the duration of the period in parenthesis (days). The duration (days) of missed collection periods due to hospitalization or other causes are listed in vertical parenthesis between collection periods. Species detected within the patient are indicated on the *y* axis, with gray denoting the species isolated from the catheter, yellow from the urine, and blue from the catheter and the urine at the given collection point. The antibiotics prescribed over the collection period are indicated at the top of each column. Immunohistochemistry staining of formalin-fixed catheter portion incubated with the indicated primary antibodies for selected catheters 132-3 (**b**) and 92-2 (**e**). Growth (log CFU/mL) of clinical *E. faecalis* (EF) or *E. coli* (Ec) isolates collected at the same collection period from Patient 132 (**c**) and Patient 92 (**f**), respectively, in monocultures (SINGLE) or mixed (MIXED) cultures in AUM. Growth of prototypical strains *E. faecalis* OG1RF with *E. coli* UTI89 are included. Data are presented as the mean values with error bars indicating the standard deviation from *n* = 3 biological replicates. Comparisons conducted using a two-sided unpaired *t* test. *$P \leq 0.05$, **$P < 0.01$, ***$P < 0.001$, ****$P < 0.0001$, ns indicates not significant.

performed immunohistochemistry on the portion of the catheter that was formalin-fixed upon collection. The fixed catheters were incubated with anti-*E. faecalis* and anti-*E. coli* primary antibodies to determine the location of the bacterial species. We observed a large overlap in the signals for the detection of *E. coli* and *E. faecalis*, suggesting that the species were co-localizing on the catheters (Fig. 6b and Supplementary Fig. 10). To study the interactions between *E. faecalis* and *E. coli* that facilitate their co-existence and colocalization in the polymicrobial community, we tested the growth of these two bacteria when grown alone or in mixed cultures using a nutrient-limited artificial urine medium (AUM, see "Methods")[23]. All *E. faecalis* strains tested, including model strain OG1RF, and 12 clinical isolates collected from Patient 132 (EF132-0 - EF132-11, Supplementary Table S14) exhibited deficient bacterial growth in AUM (Fig. 6c). Unlike *E. faecalis* isolates, *E. coli* strains, including model strain UTI89 and the 12 clinical isolates collected from patient HUC132 (EC132-0 - EC132-11, Supplementary Table S14) were able to achieve robust growth in nutrient-limited AUM (Fig. 6c). Moreover, we discovered that, compared to the monoculture of *E. faecalis* in AUM, *E. faecalis* growth was significantly enhanced when cocultured with *E. coli* in AUM (>2 log CFU, *P* < 0.001, Supplementary Table S17) (Fig. 6c). In contrast, *E. coli* exhibited no distinguishable growth differences between monocultures and mixed cultures with *E. faecalis* (Fig. 6c).

To determine whether the growth augmentation in AUM extended to *E. faecalis* and *E. coli* isolates outside of Patient 132, we tested the pairwise combinations of *E. faecalis* and *E. coli* isolated from the same collection periods for Patient 92, which clustered closely with the Patient 132 collection periods in the PCoA (Figs. 5b and 6d and Supplementary Table S14). Immunohistochemistry on formalin-fixed portions of the catheter revealed colocalization of *E. faecalis* and *E. coli* on the catheters (Fig. 6e and Supplementary Fig. 11). Similar to Patient 132, we found that the *E. faecalis* isolates from Patient 92 grew poorly in AUM monoculture and that growth was augmented in coculture with *E. coli* isolates (Fig. 6f and Supplementary Table S17). Again, *E. coli* growth in AUM appeared to be unaffected by the presence or absence of *E. faecalis* co-culture (Fig. 6f). In contrast, in rich BHI medium, all 22 *E. faecalis* isolates were able to achieve robust bacterial growth in both mono and mixed cultures (Supplementary Fig. 12), demonstrating no significant growth differences between the mono and mixed cultures, similar to what was observed for the *E. coli* strains in AUM. In addition, time series experiments indicated that while *E. faecalis* exhibited deficient growth in monoculture at 12-, 24-, and 48-h time points in AUM, *E. coli* isolates were able to promote *E. faecalis* growth in mixed cultures at all time points (*P* < 0.001, Supplementary Fig. 13). *E. coli* isolates exhibited no growth differences between the mono and mixed cultures at 12-, 24-, and 48-h time points.

We also characterized two additional pairs of bacterial interactions, including *E. faecalis* and *K. pneumoniae* as well as *E. faecalis* and *P. aeruginosa*. We chose these pairs based upon the isolate abundance, co-occurrence, and longitudinal co-colonization analyses results. *E. faecalis* and *K. pneumoniae* isolates were identified with positive co-occurrence at collection period 10 (Fig. 4k), and in vitro bacterial interaction tests between these two species collected at periods 10 and 11 from Patients 92 and 100 (Supplementary Table S15) revealed that *K. pneumoniae* promoted the growth of *E. faecalis* in the mixed cultures using AUM (Supplementary Fig 14). *P. aeruginosa* was also found to co-colonize the urinary tract together with *E. faecalis* for a long period in Patient 132 (Fig. 6a), and bacterial interaction tests of these two species in Patient 132 (Supplementary Table S16) identified one pair with mutually beneficial interaction, with *P. aeruginosa* enhancing the growth of *E. faecalis* in the co-culturing system in AUM (Supplementary Fig. 15).

Together, these results suggest that mutually beneficial interactions between different bacterial species exist in the polymicrobial community to achieve enhanced bacterial growth under nutrient-limited conditions, which may contribute to the co-colonization of various species and lead to polymicrobial CAUTI[15,27]. This is consistent with our findings from the whole patient cohort with multiple bacterial strains co-colonizing the urinary tracts of long-term catheterized patients.

## Discussion

Urinary catheters are placed in 60% of critically ill patients and 5–22% of residents in long-term care facilities[28–31]. While catheterization often represents a necessary medical intervention, placement of the catheter alters the bladder environment and provides a nidus for bacterial colonization. Bacteriuria incidence increases 3–7% each day the catheter is in place, resulting in colonization for nearly all patients with long-term catheters[10,11]. Further, 25% of all sepsis isolates can be traced back to urinary pathogens[32]. Accordingly, long-term catheters are a major source for morbidity and mortality for the patients in which they reside, with some long-term care facilities reporting over 50% of bacteraemic episodes attributed to pathogens associated with CAUTI[33]. It is critical that we improve our understanding of these infections to reduce the burden of this persistent and pervasive disease[33,34].

Our study of microbial community composition of the lower urinary tract, including voided urine and catheters from 55 long-term catheterized patients that were collected approximately monthly over a year, shows that while each patient is colonized by 6–7 species over all collection periods, only 2–3 unique species were identified at a single collection period. Only 20% of samples were found to contain a single species. Among the patients included in this study, ~73% (40/55) received at least one course of antibiotics over the study enrollment period with nearly 50% (27/55) receiving more than one course of antibiotic and more than one type of antibiotic.

We observed that several patients, including Patient 82, 92, 97, 100, 104, 115, 123, 127, 130, and 133 (Supplementary Figs. 1B, 2C, F, 3C, D, 4D, 5F, 6D, and 7A, C) received antibiotics over 4 or more collection periods. Despite this recurrent antibiotic treatment, one or more bacterial species persisted on their catheters. Given the growing concern over the rising incidence of antimicrobial-resistant uropathogens, we assayed the effect of long-term antibiotic use on the antibiotic resistance profiles of the bacterial species that persisted in the lower urinary tract environment for Patient 104, a male with a spinal cord injury. Patient 104 was prescribed the combination ß-lactam antibiotic amoxicillin with the ß-lacatmase inhibitor clavulanic acid over 5/6 collection periods. Despite this antibiotic use, the catheters collected from Patient 104 were consistently colonized by the opportunistic pathogen *Serratia marcescens* (Supplementary Fig. 3D). As ß-lactam antibiotics represent one of the most prescribed classes of antibiotics, we tested the *S. marcescens* isolates from each collection period for patient 104 against a range of ß-lactam antibiotics, including amoxicilliln (penicillin derivative), ampicillin (penicillin derivative), cefazolin (first-generation cephalosporin), ceftriaxone (a third-generation cephalosporin), cefepime (a fourth-generation cephalosporin), and meropenem (carbopenam antibiotic). All 6 *S. marcescens* isolates were highly resistant to amoxicillin, ampicillin, cefazolin, and ceftriaxone. Interestingly, for cefepime and meropenem, which are used to treat highly resistant infections, the *S. marcescens* isolate from the first collection period exhibited a MIC of 0.5 and 0.25 μg/mL respectively, well below the level of clinical resistance (4 and 2 μg/mL, respectively), however the isolate from the final collection period exhibited resistances much closer to the clinical resistance thresholds at 2 and 1.5 μg/mL, respectively (Table 1)[35]. These data suggest that long-term catheterized patients may be at increased risk of developing antibiotic-resistant infections, and that further understanding of the composition of these communities may aid in the development of novel strategies to develop effective therapeutics.

While the growing prevalence of antimicrobial-resistant uropathogens represents a significant public health crisis on its own, the

**Table 1 | Minimum inhibitory concentrations (MICs) of the listed ß-lactam antibiotics against the indicated *S. marcescens* isolates from Patient 104**

| *S. marcescens* 104 | Augmentin (penicilllin, >32) | Ampicillin (penicillin, >8) | Cefazolin (1st gen, >2) | Ceftriaxone (3rd gen, >2) | Cefepime (4th gen, >4) | Meropenem (carbopenam, >2) |
|---|---|---|---|---|---|---|
| 104-0 | No effect | No effect | No effect | 16 | 0.5 | 0.25 |
| 104-1 | No effect | No effect | No effect | 32 | 3 | 1.5 |
| 104-2 | No effect | No effect | No effect | 24 | 1.5 | 1.5 |
| 104-3 | No effect | No effect | No effect | 24 | 1.5 | 1 |
| 104–4 | No effect | No effect | No effect | 32 | 3 | 1.5 |
| 104-5 | No effect | No effect | No effect | 32 | 2 | 1.5 |

The clinical resistance cutoff for each antibiotic is indicated in parentheses. MICs are listed in µg/mL.

polymicrobial nature of CAUTI makes these infections further recalcitrant to treatment with our current arsenal of antibiotics. Thus, better understanding of the polymicrobial community composition and the mechanisms that drive positive co-occurrences will allow for targeted treatments that aim to collapse the community structure, rendering the individual uropathogens more susceptible to treatment.

*E. faecalis, E. coli, P. mirabilis, K. pneumoniae*, and *P. aeruginosa* were amongst the most common species colonizing the catheters in this study, which is similar to a previous study where the highest colonizers from weekly urine samples in a long-term catheterized patient cohort included *E. faecalis, P. mirabilis, P. stuartii*, and *S. aureus*[15]. Armbruster et al. also identified highly concordant species pairs in the urine, including *P. mirabilis* and *P. stuartii*, which is consistent with the positive associations detected between Proteus and Providencia described across several collection periods here and *P. mirabilis* and *M. morganii*, which also appeared during multiple collection periods between Proteus and Morganella in our study[15]. They also identified concordant species that were not identified across any time points in our study, including *P. stuartii* and *P. mirabilis* with *E. faecalis*. Conversely, our study identified positive co-occurrences between the highly sampled Klebsiella and Pseudomonas genera as well Klebsiella and Enterococcus that were not identified in Armbruster et al. These differences could be attributable to differences in study design, where Armbruster et al. analyzed weekly urine samples from 17 long-term catheterized patients while we received both catheter and urine samples approximately monthly from 55 long-term catheterized patients.

It is interesting to note that the Gram-positive staphylococci represent the genus with the most negative co-occurrences and are the most frequently monomicrobial samples. *S. aureus* has been shown to elicit a robust pro-inflammatory response in the murine bladder, which could act to reduce colonization of competing species[24]. These negative interactions were best exemplified by Staphylococcus and Enterococcus in our long-term catheterized patient cohort, which had negative co-occurrences in 7/12 collection periods in our study. In contrast to these results, a recent study by Ch'ng et al. has shown positive interactions between *Staphylococcus aureus* and *Enterococcus faecalis* that result in augmented biofilm production and facilitated *E. faecalis* growth through a *S. aureus*-derived heme cross-feeding mechanism[36]. Ch'ng et al. analyzed isolates from various body sites, whereas our study was limited to isolates in the urinary tract. Ch'ng et al. showed that *S. aureus* USA300LAC had drastically heterogeneous responses in augmenting biofilm biomass when cocultured with a diverse panel of *E. faecalis* isolates, including bloodstream, wound, UTI, and gastrointestinal isolates, where only 6/32 *E. faecalis* isolates showed increased biomass in co-culture, suggesting the environment and other factors may be important for dictating the nature of the interaction between staphylococcal and enterococcal species. Consistent with the results presented here, among the *E. faecalis* isolates tested in the previous study, those isolated from UTIs did not show a statistically significant augmentation in dual-species biofilms when cocultured with *S. aureus* USA300LAC[36]. Together these data suggest

that the environment in which these species co-occur dictates how they interact with one another.

One limitation of this study is the use of a culture-based approach, wherein organisms that grow well under the selected laboratory conditions, namely in rich undefined media at 37 °C, will grow and be detected in subsequent 16 S sequencing whereas fastidious and anaerobic species will be lost due to their inability to grow under these conditions. Thus, a culture-based approach limits the species that can be included in the downstream co-occurrence analysis. While newer sequencing applications can be used to sequence directly from catheter and urine samples, circumventing the limitations noted above, we chose to use a culture-based approach as that is currently the gold standard in clinical microbiology laboratories. Future studies are necessary to bridge the gap between standard clinical techniques for microbial detection and newer sequencing-based approaches that are not currently used in the clinical setting. Also necessary are studies of the cytokines and host factors present in the urine of long-term catheterized patients, which will allow for better understanding of the immune responses associated with certain polymicrobial community compositions. The microbial positive associations detected in our study represent those present in human urinary tracts of long-term catheterized patients spanning many subsequent re-catheterization events, warranting further investigation to the molecular drivers of these interactions for insights to perturb the formation and stability of problematic polymicrobial communities.

## Methods
### Study approval
For all patients included in this study, informed consent was obtained. As previously described, the urines and catheters were collected upon the clinical decision to remove the catheter for standard-of-care[25,37]. This study was approved by the Washington University School of Medicine (WUSM) Internal Review Board (approval #201410058) and performed in accordance with WUSM's ethical standards and the 1964 Helsinki declaration and its later amendments.

### Statistics
For the co-occurrence analysis, all *P* values were calculated using the cooccur package in R[26]. For the PCoA, the dissimilarity matrix was generated using the "vegdist" function from the Vegan package in R using the default settings which was passed to the built in "cmdscale" function[38]. The species scores were generated using the 'add.spec-scores' function from the BiodiversityR library[39].

Comparisons of bacterial growth (measured in colony-forming units, CFUs) from mono- and mixed cultures were conducted using an unpaired *t* test in GraphPad Prism 9.0 (GraphPad software).

### General bacteriology
All strains were grown in brain heart infusion (BHI) or Luria-Bertani (LB) media restreaked onto BHI+agar or LB+agar plates unless stated otherwise and grown overnight at 37 °C statically.

## Sample preparation

As previously described[24,25], upon receiving the samples at the lab, 0.5 cm of tip of the catheter was cut and placed into BHI liquid media and samples were grown overnight at 37 °C. The resulting overnight cultures were then plated on BHI-agar and grown overnight at 37 °C. The remaining catheter was split into two pieces and one half was fixed in formalin for immunohistochemistry. A small portion of the urine (10 µL) was plated onto BHI-agar and grown overnight at 37 °C and the remaining sample was aliquoted into 50 mL conical tubes and stored at −20 °C. Colonies were distinguished based on size, morphology, and color and 1–9 single colonies per sample were selected for DNA extraction. 16 S sequencing was performed as previously described[24,25].

## Bioinformatics

All basic plots were generated in R v4.2.1 using Rstudio v2022.12.0 + 353. Repeated species identifications from the same collection period were condensed into one entry and the source of the isolate (urine, catheter, or both) was noted. For the co-occurrence analysis, a matrix of species occurrence by site was generated using the Source Data as an input stratified by time point. The co-occurrence analysis was completed using the package co-occur in R[26] with default settings, except where the threshold option was switched to 'false' to include the co-occurrences between less abundant species.

## Immunohistochemistry on human catheters

The formalin-fixed catheters were prepared as described previously[25,37]. Briefly, catheter samples were washed three times with PBS and subsequently blocked with PBS containing 1.5% BSA and 0.1% sodium azide overnight at 4 °C. The blocked catheters were then washed three times with PBS-T and incubated with Enterococcus polyclonal antibody (PA1-73120, Invitrogen, validation statement for immunofluorescence application available at: https://www.thermofisher.com/antibody/product/Enterococcus-Antibody-Polyclonal/PA1-73120) and *E. coli* serotype-O/K Polyclonoal Antibody (PA1-73032, Invitrogen, validation statement for immunofluorescence application available at https://www.thermofisher.com/antibody/product/E-coli-serotype-O-K-Antibody-Polyclonal/PA1-73032) at 1:400 dilutions in dilution buffer (PBS-T, 0.1% w/v BSA, 0.5% methyl alpha-D-mannopyranoside) for two hours at room temperature. The catheters were then washed and incubated with IRDye 680LT Donkey anti-Rabbit (LI-COR, 926-68023) and IRDye 800CW Donkey anti Goat (LI-COR, 926-32214) antibodies (1:10,000 dilution in dilution buffer) for one hour at room temperature. The catheters were then washed three times in PBS-T and imaged using the Odyssey Imaging System (LI-COR Biosciences) and merged in Adobe Photoshop. The control portion of the catheter was not incubated with primary antibody.

## Clinical isolate mono- and co-culturing in AUM and rich BHI media

Twenty-two pairs of *E. faecalis* and *E. coli*, including one pair of model strains (*E. faecalis* OG1RF and *E. coli* UTI89), 9 pairs of clinical isolates from patient HUC92, and 12 pairs of clinical isolates from patient HUC132, were examined in a pairwise manner for studying the interaction between these two bacterial species (Supplementary Table S14). The pair of OG1RF and UTI89 was used as the example to illustrate the experiment design as follows. OG1RF and UTI89 strains were streaked out from frozen stocks on brain heart infusion (BHI) and Luria-Bertani (LB) agar plates, respectively, and grown overnight at 37 °C statically. Next, OG1RF and UTI89 cultures were grown from multiple colonies in BHI and LB media, respectively, overnight at 37 °C statically, adjusted to OD = 1.0 using fresh media, and washed with PBS three times[40–42]. Three types of bacterial inoculums, OG1RF, UTI89, or OG1RF plus UTI89 (OG1RF + UTI898) were inoculated into AUM or rich BHI media at 1:1000 dilution (OD = 0.001), and 200 µL cultures were added into 96-well plates to grow for 48 hours at 37 °C statically. At 12, 24, or 48 h,

OG1RF, UTI89, and OG1RF + UTI89 cultures were plated on LB + Kan50 (Kanamycin = 50 µg/mL), LB + Vanco6 (Vancomycin = 6 µg/mL), and both LB + Kan50 and LB + Vanco6 agar plates, respectively, for CFU enumerations as CFU/mL[18,41,43]. For *E. faecalis* and *E. coli* strains tested in this study, *E. faecalis* strains were resistant to Kan50 but sensitive to Vanco6, while *E. coli* strains were resistant to Vanco6 but sensitive to Kan50, therefore Kan50 and Vanco6 antibiotic agar plates were able to differentiate between *E. faecalis* and *E. coli* strains in the mixed cultures. The 9 and 12 pairs of *E. faecalis* and *E. coli* clinical isolates from patients HUC92 and HUC132, respectively, were tested in the same manner to measure the bacterial growths in mono or mixed cultures for investigating interactions between *E. faecalis* and *E. coli* in the polymicrobial community. Additionally, four pairs of *E. faecalis* and *K. pneumoniae* strains from Patient 92 and 100 (Supplementary Table S15), and two pairs of *E. faecalis* and *P. aeruginosa* strains from Patient 132 (Supplementary Table S16), were also selected for 48-h bacterial interaction studies using the same protocol as described above. Antibiotic plates were used for differentiating bacterial species in the mixed cultures. For characterizing *E. faecalis* and *K. pneumoniae* interactions, *E. faecalis* strains were resistant to Kan50 but sensitive to Vanco6, while *K. pneumoniae* strains were resistant to Vanco6 but sensitive to Kan50, therefore Kan50 and Vanco6 antibiotic agar plates were able to differentiate between these two species in the mixed cultures. For characterizing *E. faecalis* and *P. aeruginosa* interactions, *P. aeruginosa* strains were resistant to Vanco6, while *E. faecalis* strains were sensitive to Vanco6, therefore Vanco6 antibiotic plates were used for counting *P. aeruginosa* CFU in the co-cocultures and *E. faecalis* CFU was determined by total CFU minus *P. aeruginosa* CFU.

## Antibiotic susceptibility testing

Isolates were streaked from frozen stocks onto agar plates and grown overnight at 37 °C. A single colony was placed in liquid media for overnight growth at 37 °C. Overnight cultures were normalized to an $OD_{600}$ of 2.0, and 100 µL of normalized culture was spread onto agar plates using sterile glass beads. Antibiotic test strips from Licofilchem™ MTS™ for Ampicillin (22-777-657), Amoxicillin-Clavulanic Acid (22-778-044), Cefazolin (22-778-041), Cefepime (22-777-915), Ceftriaxone (22-777-915), and Meropenem (22-777-858) were placed onto the bacteria-agar plates using sterile forceps, with a maximum of two inverted antibiotic strips per plate. Serial dilutions of the culture inoculums were plated for initial colony-forming units and all plates were incubated at 37 °C. MICs and initial CFUs were enumerated and recorded the following morning.

## Reporting summary

Further information on research design is available in the Nature Portfolio Reporting Summary linked to this article.

## Data availability

Sequencing files for isolates sequenced after January 14, 2020 have been deposited at the NCBI Sequence Read archive under BioProject ID PRJNA956906. Despite exhaustive efforts to locate the original sequencing prior to this date, the authors were unable to locate the original files. All analyses have been performed on the subset of data for which the sequencing data exists (see Supplementary Figs. 8 and 9) and the conclusions, including the average species identified per patient, species identified per collection period, and relative species abundance were the same. Source data are provided with this paper.

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

## Acknowledgements

This work was supported by the National Institutes of Health RO1DK51406 (S.J.H. and M.G.C.) and KO1 DK128381-O1A1 (J.N.W). T.M.N. was supported by the W.M. Keck Postdoctoral Fellowship and by the National Institute of Allergy and Infectious Diseases of the National Institutes of Health under Award Number T32AI007172. The authors thank the lab of Dr. Matthew Chapman for their feedback and support in manuscript preparation. We also thank all study participants for their sample donations.

## Author contributions

T.M.N., Z.Z., A.L.F.M., J.N.W., K.W.D., A.D., M.G.C., and S.J.H. designed the research studies, T.M.N., Z.Z., C.L.P.O., J.S.P., E.L., K.K, K.B., A.K., A.L.F.M., J.N.W., and D.W. conducted experiments and acquired data, T.M.N., Z.Z., and D.W. analyzed the data, T.M.N. and Z.Z. wrote the first draft of the manuscript, and all authors contributed to the editing of the manuscript.

## Competing interests

The authors declare no competing interests.
