## [Peer Review File · Nature Communications]

REVIEWER COMMENTS

Reviewer #1 (Remarks to the Author):

In this study, Nye and Zou et al examined the microbes present in longitudinal urine and catheter samples to identify positive and negative associations. They report that 80% of 366 urine and catheter samples were polymicrobial, *E. faecalis* was the most common species overall, and a positive association exists between *E. faecalis* and *E. coli*. This positive association was further explored in vitro using artificial urine medium by testing two reference strains and paired longitudinal clinical isolates, through which *E. coli* was found to augment growth of *E. faecalis*.

There is growing interest regarding the composition of polymicrobial urine specimens due to the impact on antimicrobial treatment efficacy, and few studies include repeat assessment from a given patient. This well-written manuscript therefore adds to our current understanding of the potential impact of microbial co-occurrence, including the ability of some species to augment growth or survival of others. While many of the conclusions are well-justified, further clarification is needed regarding certain details of the patient study and the co-occurrence analyses, as well as discrepancies between the main figures and the supporting data. Further comment on similar studies is also warranted. Specific comments are as follows:

1. In general, the methods are lacking important details pertaining to the study design and subjects. What were the inclusion and exclusion criteria for catheter and urine sample collection? For how long had the patient required an indwelling catheter? This is an important consideration, as the microbiology may be very different for a patient who has only had a catheter for a few weeks compared to someone with an indwelling catheter for a few decades. Further, line 132 listed “temporary post-operative” as an indication for catheter use, which does not suggest long-term catheterization. It would also be helpful to include how recently each patient had been exposed to antibiotics prior to sample collection, and if the antibiotics were for a urinary tract infection.
2. For concordance analysis, the `cooccur` package in R assumes the distribution of each species to be random and independent. Repeated samples from the same individual are not independent, which needs to be accounted for in the statistical analysis.
3. Co-occurrence of microbes in catheter urine specimens was also recently assessed in longitudinal samples from patients with long term catheters (PMID 34473649, reference 15 of this study). That study observed numerous positive associations that were also observed here, including between *E. faecalis* and *E. coli*, *P. mirabilis* and *P. stuartii*, *P. mirabilis* and *M. morgani*, along with several other associations

not detected here, such as *P. mirabilis* with *E. faecalis*. Further commentary on the similarities and differences between these studies and their findings is warranted.

4. The methods indicate that catheters cultures were obtained by placing the catheter tip in BHI broth and incubating at 37C overnight prior to plating on BHI agar. However, this outgrowth in BHI is likely to introduce substantial bias into the microbial composition of each sample. The potential impact on the identified positive and negative relationships warrants discussion as a limitation of the study design.

5. Regarding the co-culture of *E. coli* and *E. faecalis*, interactions between these species were previously shown to augment *E. coli* growth under certain conditions (PMID 2773664). It is therefore interesting that only the growth of *E. faecalis* was augmented in artificial urine. Considering that these experiments assessed viability at a single time point (48 hours), might co-culture enhance viability of either species at earlier time points, or have a differential temporal effect? This would be beneficial to show for at least one representative pair from each panel of isolates, along with the control strains.

6. Based on the methods, it would appear that both a urine sample and a catheter sample were cultured for each collection. How concordant were the genera between the urine and catheter samples from a given individual? Did the concordance analysis include genera detected from any source at a given timepoint, or only a specific source (for instance, if *E. coli* was only present on the catheter and *K. pneumoniae* was only present in the urine, as in Sample 9 from Figure 5A, was this considered concordant?)

7. There is some confusion regarding the sequencing data and the supplemental figures vs the main figures. Lines 342-348 indicate that the sequencing files are only available for isolates sequenced after Jan 2020, and that only this subset was used for analyses. However, it is unclear what portion of the 366 catheter and urine specimens are lacking sequencing data, although I suspect this may be why there are 55 patient IDs in figure 1B and only 26 in supporting figure 1. Further clarification is needed, particularly how the data from the supplemental figures align with the main figures. How many samples are missing sequencing data for each patient ID? The data presented in supporting figures 1 and 2 are also not referenced in the main text.

8. More information is needed to interpret Figure 2B. The legend indicates that these are genera detected over all study visits, but this does not provide information about temporal detection within an individual, or abundance, or whether the detected genera was from the catheter vs the urine specimen (same issue with Figure 3B).

9. For the temporal data in Figure 2C, how are sparse data handled? For instance, *Achromobacter* has a count of 2 from panel A, which appear to be from sample collection number 0 and 1. Was this from a

single participant, or was this genus found in two participants, one at visit 0 and one at visit 1? If from a single patient and only at 2 timepoints, how can temporal frequency be inferred? The results also state that frequency was assessed through $t=11$, but it is unclear how species from patients with fewer collections were handled for frequency calculations. For instance, were Enterococcal, Staphylococcal, Proteus, and Pseudomonas species truly present more frequently during early collection periods, or did the participants who had these species just have fewer overall collections?

10. Lines 164-166 indicate a median of 6 species (range 1-16) per patient, but lines 178-181 indicate a median of 2 (range 1-8) for a given individual at a given collection, indicating that only one or two catheter samples per patient had a high number of species. Were there events flanking the high polymicrobial samples that might explain the change in microbial diversity?

11. Lines 314-317 state a positive co-occurrence between Ef and Pa in AUM, but the data are not shown in the manuscript or supporting data

Reviewer #2 (Remarks to the Author):

In the current manuscript the authors collected a large number of urine and catheter samples from long-term catheterized patients over a year. They isolated the CAUTI-causing pathogens from these samples in a culture-dependent manner by subculturing urine and catheters in a single culture media under aerobic conditions. The authors showed that the majority of CAUTIs were polymicrobial, with *E. faecalis* being the most common species. They also evaluated the different bacterial co-occurrences, which revealed that *E. faecalis* is positively co-associated with *E. coli* and other gram-negative species, but negatively co-associated with gram positive species. They went on studying the interactions of *E. faecalis* and *E. coli* in vitro using artificial urine media, showing that *E. coli* promotes growth of *E. faecalis*.

The main strength of this manuscript lies in the extensive collection of samples, which adds on to the existing literature that describes the composition of polymicrobial communities in CAUTI patients. It also provides insight into bacterial co-occurrences in CAUTI, which is of interest to microbiologists studying polymicrobial infections. However, the current study design provides little insight on several unanswered questions in the field of CAUTI, such as i) the evolution of polymicrobial communities in the host over time, ii) the impact of antibiotic treatment on the polymicrobial communities, iii) the immune response to polymicrobial communities and iv) the origin of infecting pathogens.

Major comments

1) The major weakness of the current study is the culture-dependent isolation of the CAUTI-causing pathogens. Bacteria were isolated after subculturing or plating the samples in only one media (BHI) under aerobic conditions. The use of only one culture condition might have led to the underestimation of the species number comprising the polymicrobial communities in CAUTI. Moreover, by subculturing the urine samples, information on the relative abundance of each species at different time-points during catheterization is lost.

The application of a culture-independent approach, such as metagenomics would have provided a more precise description of the polymicrobial community composition in CAUTI patients, as well as the relative abundance of each species in these communities over time.

After reading the methods section of the current manuscript, I understand that the authors have collected catheter samples for immunohistochemistry and DNA isolation. The analysis and inclusion of these samples either with FISH or genome sequencing would add a lot of value to this manuscript because it will reveal the microbial diversity in a culture-independent manner and provide insight into the relative abundance of each species. I recommend that the authors revise their study design accordingly.

2) I understand that the same species were isolated on more than one occurrence from the same patient. Other studies have shown that bacteria can evolve in the host, especially under antibiotic pressure. Given that the authors already have the bacterial samples in their possession, it would add great value to this manuscript if they showed whether any of the infecting species evolved over time in the presence/absence of antibiotic treatment or whether different strains of the same species were present in patients.

3) In Fig2C, the authors are showing a summary of the isolation frequency of different species over the different collection time-points. However, it is unclear whether the sample collection interval (eg. Number of days post catheterization) was similar for all recruited patients. Moreover, the authors summarize data from a patient group with different underlying pathologies, age, sex and antibiotic treatment, which makes difficult to draw any conclusion. Did the authors find any associations between sex, age, underlying pathologies, antibiotic treatment and polymicrobial community composition? Please include this information in the manuscript if available. Moreover, I suggest to the authors to show the polymicrobial composition for each patient individually over time, as well as include information such as the onset of antibiotic treatment and the class of antibiotic used.

4) I understand that the authors have stored urine samples collected, which would be a great opportunity to investigate how polymicrobial communities influence the immune response. Did they perform any urine analysis to quantify host factors, such as cytokines, and if so, can they include any such findings in their study?

Minor comments

It is unclear what the counts on the Y axes of figure 3A and B correspond to. Kindly clarify.

Reviewer #3 (Remarks to the Author):

Microbial co-occurrences on catheters from long-term catheterized patients

General comments: In this manuscript, the authors obtained access to urinary catheters from 55 human patients that were catheterized long-term. Extended follow-up was performed on these patients that often had catheters replaced on a monthly basis, leading to on average (~ 6.7 samples/patient over the course of the study). Portions of the catheter were inoculated into BHI and then plated on BHI agar for colony identification. Urine samples from the same patient were also grown on BHI agar for colony identification. In total 366 samples were evaluated. ~ 80% of the samples were polymicrobial in nature. The authors did an analysis of co-occurrences in the polymicrobial interactions, and found 19 positive associations and 20 negative associations among the microbial cohorts.

The major take home of this study is that the observations made about positive and negative associations among microbes is that these interactions are very much context dependent, and refute recent studies where media and culture dynamics don't often resemble the in vivo micro-environment. In particular, they focused on *Enterococcus faecalis* as this bacteria was present in 187/366 samples. The next closest species in abundance was *Escherichia coli* in 102/366 samples. Noteworthy was the negative association displayed by *E. faecalis* with other Gram-positive species and the fungal pathogen, *Candida albicans*, and the positive association with predominant Gram-negative species, including *E. coli*, *Enterobacter* sp., *Klebsiella oxytoca* and *Pseudomonas aeruginosa*.

Focusing on the positive association between *E. faecalis* and *E. coli*, the authors performed mono and co-culture analysis in Artificial Urine Media (AUM). In AUM, *E. faecalis* growth was augmented 2 orders of magnitude in the presence of *E. coli*.

Overall, the study provides a clinical snapshot of microbial interactions in the context of urinary tract infection during long-term catheterization. Although no mechanistic insights were provided as to why certain microbial interactions occurred, the study nevertheless provides a roadmap for future studies.

The major critique of the study is why the authors chose culture-based approaches to determine species composition growing on the catheter and not a direct sequencing approach. It would appear that the culture-based approach may exclude lower abundant, but important UTI community members that may contribute to the course of infection.

Specific comments:

Page 5, lines 10-12: This sentence seems redundant. 80% are polymicrobial with the majority containing 2-3 detected bacterial species. By definition, polymicrobial indicates 2 or more. Please consider restating this sentence.

Page 6, lines 6-7. It is a minor point, but when it comes to UTI and catheterization, gender identification seems to take a backseat to biological sex. If the patients are male or female, it seems appropriate to not mention how they identify for a biological variable.

Page 7, line 9 as well as other places in the manuscript, the use of enterococcal or staphylococcal as adjectives should not be capitalized. A more appropriate way to deal with this sentence and to be consistent with *Proteus* and *Pseudomonas* species would be to state it as *Enterococcus*, *Staphylococcus*, *Proteus*, and *Pseudomonas* species.

In the methods section under sample preparation, it is mentioned that the remaining portions of the catheter, were split evenly with one half used for immunohistochemistry and the other half stored in DNA shield. No further mention is made of performing immunohistochemistry or genomic DNA extractions from the catheter. Since no downstream analysis was performed on these samples, it seems appropriate to omit this detail.

REVIEWER COMMENTS

Reviewer #1 (Remarks to the Author):

In this study, Nye and Zou et al examined the microbes present in longitudinal urine and catheter samples to identify positive and negative associations. They report that 80% of 366 urine and catheter samples were polymicrobial, *E. faecalis* was the most common species overall, and a positive association exists between *E. faecalis* and *E. coli*. This positive association was further explored in vitro using artificial urine medium by testing two reference strains and paired longitudinal clinical isolates, through which *E. coli* was found to augment growth of *E. faecalis*.

There is growing interest regarding the composition of polymicrobial urine specimens due to the impact on antimicrobial treatment efficacy, and few studies include repeat assessment from a given patient. This well-written manuscript therefore adds to our current understanding of the potential impact of microbial co-occurrence, including the ability of some species to augment growth or survival of others. While many of the conclusions are well-justified, further clarification is needed regarding certain details of the patient study and the co-occurrence analyses, as well as discrepancies between the main figures and the supporting data. Further comment on similar studies is also warranted. Specific comments are as follows:

1. In general, the methods are lacking important details pertaining to the study design and subjects. What were the inclusion and exclusion criteria for catheter and urine sample collection? For how long had the patient required an indwelling catheter? This is an important consideration, as the microbiology may be very different for a patient who has only had a catheter for a few weeks compared to someone with an indwelling catheter for a few decades. Further, line 132 listed “temporary post-operative” as an indication for catheter use, which does not suggest long-term catheterization. It would also be helpful to include how recently each patient had been exposed to antibiotics prior to sample collection, and if the antibiotics were for a urinary tract infection.

We thank the Reviewer for bringing this to our attention. We have updated the main text to include the inclusion and exclusion criteria (lines 115-118) and reported the average catheter dwell times (line 114) for the samples collected in the study. We did not have access to the patient data prior to the study to know how long the patient had required an indwelling catheter prior to enrollment. For the indicated patient (Patient 96), though only one sample was collected where *E. faecalis* and *E. coli* were detected on the catheter and in the urine, this patient had an indwelling catheter for 56 days and had surgery that made catheterization no longer necessary. We now note this in the manuscript in line 112.

We also created **Supplementary Figure 1A-OO**, which provides detailed information on the catheter dwell times, missed collection periods, antibiotics prescribed, species identification, and the source of the collection (catheter, urine, or both) for each collection period by patient for all patients with greater than one collection period and two or more species identified.

2. For concordance analysis, the cooccur package in R assumes the distribution of each species to be random and independent. Repeated samples from the same individual are not independent, which needs to be accounted for in the statistical analysis.

In order to address this error, we have performed the co-occurrence analysis across the sites (urine and catheter) collected from all patients at a given collection period (period 1-12), ensuring that the samples are independent. This analysis now replaces the previous Figure 4. We performed the analysis at the genus level to account for any errors in species-level calls via the 16S identification method. Importantly, we found that *Staphylococcus* maintained several negative co-occurrences with *Pseudomonas*, *Enterococcus*, and *Klebsiella*. We also identified several positive co-occurring genera across various time points and compared this with findings from other studies in the discussion.

3. Co-occurrence of microbes in catheter urine specimens was also recently assessed in longitudinal samples from patients with long term catheters (PMID 34473649, reference 15 of this study). That study observed numerous positive associations that were also observed here, including between *E. faecalis* and *E. coli*, *P. mirabilis* and *P. stuartii*, *P. mirabilis* and *M. morgani*, along with several other associations not detected here, such as *P. mirabilis* with *E. faecalis*. Further commentary on the similarities and differences between these studies and their findings is warranted.

We now include a section in the discussion comparing the findings between PMID 34473649 and our study in lines 330-346. In addition to highlighting the similar co-occurrences that were detected we also note that there were several differences in study design that may explain differences between the studies.

4. The methods indicate that catheter cultures were obtained by placing the catheter tip in BHI broth and incubating at 37C overnight prior to plating on BHI agar. However, this outgrowth in BHI is likely to introduce substantial bias into the microbial composition of each sample. The potential impact on the identified positive and negative relationships warrants discussion as a limitation of the study design.

We now discuss this limitation to our study design in the discussion section in lines 370-383. We note that while newer methods are available for sequencing directly from catheter and urine samples, which can be used to circumvent the noted limitations, these are not currently regularly available in the clinical setting. Thus, we chose to focus our study on polymicrobial communities that would be routinely detected in the clinical setting.

5. Regarding the co-culture of *E. coli* and *E. faecalis*, interactions between these species were previously shown to augment *E. coli* growth under certain conditions (PMID 2773664). It is therefore interesting that only the growth of *E. faecalis* was augmented in artificial urine. Considering that these experiments assessed viability at a single time point (48 hours), might co-culture enhance viability of either species at earlier time points, or have a differential

temporal effect? This would be beneficial to show for at least one representative pair from each panel of isolates, along with the control strains.

To address this point, we grew *E. faecalis* monoculture in AUM and in co-culture with *E. coli* for prototypical strains OG1RF and UTI89, respectively, as well as the clinical *E. faecalis* and *E. coli* isolates from the first collection period for patients 92 and 132 at 12-,24-, and 48-hour time points. Consistent with our previous results, we found that *E. faecalis* grew poorly in monoculture at all time points and that growth was augmented by mixed culture with *E. coli*. Further, no growth difference was observed for *E.coli* isolates between mono and mixed cultures at 12-,24-, and 48-hour time points. This data is now shown in **Supplementary Figure 7**.

Additionally, in PMID 27736645, the authors grew the strains in very different media, specifically BHI for *E. faecalis* (which has robust growth without *E. coli*), LB for *E. coli*, and the strains were co-cultured in TSB with 10% glucose with iron depletion achieved by the addition of 2,2'-dipyridyl. Thus, the growth conditions differed dramatically from that of artificial urine media presented here and likely explain the differences between our studies.

6. Based on the methods, it would appear that both a urine sample and a catheter sample were cultured for each collection. How concordant were the genera between the urine and catheter samples from a given individual? Did the concordance analysis include genera detected from any source at a given timepoint, or only a specific source (for instance, if *E. coli* was only present on the catheter and *K. pneumoniae* was only present in the urine, as in Sample 9 from Figure 5A, was this considered concordant?)

To analyze the concordance between genera detected in the urine and the catheter samples, we plotted the source (where yellow=urine, gray=catheter, and blue=both) of each identified genus over all collection periods in **Figure 2C**. Moreover, the source of species isolation and the concordance between urine and catheter samples at the single patient level is now available in **Supplementary Figure 1 A-AO**.

The co-occurrence analysis in the main text included species detected from any source at a given time point/collection period as we sought to determine co-occurrences within the lower urinary tracts of the patient cohort, which would include both the catheter and urine samples. This point has now been clarified in the text.

7. There is some confusion regarding the sequencing data and the supplemental figures vs the main figures. Lines 342-348 indicate that the sequencing files are only available for isolates sequenced after Jan 2020, and that only this subset was used for analyses. However, it is unclear what portion of the 366 catheter and urine specimens are lacking sequencing data, although I suspect this may be why there are 55 patient IDs in figure 1B and only 26 in supporting figure 1. Further clarification is needed, particularly how the data from the supplemental figures align with the main figures. How many samples are missing sequencing data for each patient ID? The data presented in supporting figures 1 and 2 are also not referenced in the main text.

Supplementary Figures 2 and 3 (formerly Figures 1 and 2) are now referenced in the main text. To make the comparison clearer, we have indicated in the Supplementary that the distribution of species occurrences in the dataset for which sequencing reads were recovered (Fig S2A) is highly similar to the whole patient dataset (Fig 3A). We also provide comparison of the species detected by patient for the sequencing read dataset (Fig S2B) and the whole dataset (Fig S2C, formerly figure 3B) for direct comparison of the species detected per patient for the dataset.

8. More information is needed to interpret Figure 2B. The legend indicates that these are genera detected over all study visits, but this does not provide information about temporal detection within an individual, or abundance, or whether the detected genera was from the catheter vs the urine specimen (same issue with Figure 3B).

To address this, we have added **Supplementary Figure 1 A-OO**, which provides detailed information on the catheter dwell times, missed collection periods, antibiotics prescribed, species identification, and the source of the collection (catheter, urine, or both) for each collection period by patient. Thus, while Figures 2B and S2C (3B is now presented in Fig S2C) are meant to provide a summary of all genera/species detected in each patient, providing a snapshot of which genera/species were detected and how they varied by patients, detailed information about the temporal detection within an individual and the source of the isolate (catheter, urine, both) can now be found in the Supplementary. With regard to abundance, we can only provide data on the presence/absence per visit using a culture-based approach, which is why the data are presented as count data for a given patient.

9. For the temporal data in Figure 2C, how are sparse data handled? For instance, *Achromobacter* has a count of 2 from panel A, which appear to be from sample collection number 0 and 1. Was this from a single participant, or was this genus found in two participants, one at visit 0 and one at visit 1? If from a single patient and only at 2 timepoints, how can temporal frequency be inferred? The results also state that frequency was assessed through $t=11$, but it is unclear how species from patients with fewer collections were handled for frequency calculations. For instance, were *Enterococcal*, *Staphylococcal*, *Proteus*, and *Pseudomonas* species truly present more frequently during early collection periods, or did the participants who had these species just have fewer overall collections?

To address the Reviewer's concerns, we have limited our analysis to patient data that include at least 75% of the total collection periods (9/12 collections +) and present this data as the new panel in Figure 2B.

10. Lines 164-166 indicate a median of 6 species (range 1-16) per patient, but lines 178-181 indicate a median of 2 (range 1-8) for a given individual at a given collection, indicating that only one or two catheter samples per patient had a high number of species. Were there events flanking the high polymicrobial samples that might explain the change in microbial diversity?

This is a very interesting question. Patient 99 accounts for the largest number of species per patient (16) as well as the largest number of species identified per collection period (8). During the first collection period there were only four species detected between the catheter and urine samples and during the time the catheter was indwelling the patient was prescribed a course of ceftazolin, which is a first-generation cephalosporin administered intravenously. Interestingly we see an increase to 6 species detected in the third collection period, during which the patient was prescribed Keflex (cephalexin, another first-generation cephalosporin that can be administered orally). The collection period after the prescription of Keflex showed the maximum 8 species detected. This is only a single example, and therefore no definitive conclusions can be drawn, but it is interesting to note that the maximum species detected per collection period in this study occurred subsequent to antibiotic administration.

11. Lines 314-317 state a positive co-occurrence between Ef and Pa in AUM, but the data are not shown in the manuscript or supporting data

The text has been removed from the updated version of the manuscript.

Reviewer #2 (Remarks to the Author):

In the current manuscript the authors collected a large number of urine and catheter samples from long-term catheterized patients over a year. They isolated the CAUTI-causing pathogens from these samples in a culture-dependent manner by subculturing urine and catheters in a single culture media under aerobic conditions. The authors showed that the majority of CAUTIs were polymicrobial, with *E. faecalis* being the most common species. They also evaluated the different bacterial co-occurrences, which revealed that *E. faecalis* is positively co-associated with *E. coli* and other gram-negative species, but negatively co-associated with gram positive species. They went on studying the interactions of *E. faecalis* and *E. coli* in vitro using artificial urine media, showing that *E. coli* promotes growth of *E. faecalis*.

The main strength of this manuscript lies in the extensive collection of samples, which adds on to the existing literature that describes the composition of polymicrobial communities in CAUTI patients. It also provides insight into bacterial co-occurrences in CAUTI, which is of interest to microbiologists studying polymicrobial infections. However, the current study design provides little insight on several unanswered questions in the field of CAUTI, such as i) the evolution of polymicrobial communities in the host over time, ii) the impact of antibiotic treatment on the

polymicrobial communities, iii) the immune response to polymicrobial communities and iv) the origin of infecting pathogens.

Responses to the points raised in i)-iii) are addressed below. Although we agree that point iv) (the origin of infection pathogens) is a very interesting research question, it is outside the scope of this study and would be very difficult to assess via an analysis of polymicrobial communities after catheter removal.

Major comments

1) The major weakness of the current study is the culture-dependent isolation of the CAUTI-causing pathogens. Bacteria were isolated after subculturing or plating the samples in only one media (BHI) under aerobic conditions. The use of only one culture condition might have led to the underestimation of the species number comprising the polymicrobial communities in CAUTI. Moreover, by subculturing the urine samples, information on the relative abundance of each species at different time-points during catheterization is lost.

The application of a culture-independent approach, such as metagenomics would have provided a more precise description of the polymicrobial community composition in CAUTI patients, as well as the relative abundance of each species in these communities over time.

After reading the methods section of the current manuscript, I understand that the authors have collected catheter samples for immunohistochemistry and DNA isolation. The analysis and inclusion of these samples either with FISH or genome sequencing would add a lot of value to this manuscript because it will reveal the microbial diversity in a culture-independent manner and provide insight into the relative abundance of each species. I recommend that the authors revise their study design accordingly.

We agree with the Reviewer that a culture-independent metagenomic approach would have provided a more precise description of the long-term catheterized patients' polymicrobial communities. However, our current study was designed to reflect the practices currently used as the gold standard in clinical microbiology labs, to be reflective of the information that could be gathered and subsequently applied in a hospital setting, where metagenomic sequencing is not routine. Additionally, at the time of the initial study design, patients were not consented for metagenomic sequencing from urine or catheter samples, which would contain host genomic DNA and therefore be subject to new IRB approval. Although the samples were saved in DNA/RNA shield, the purpose of these samples as detailed in the IRB was for the performance of future transcriptomic studies, which are more amenable to host nucleic acid depletion techniques than DNA samples. We have been working to update the study design and the IRB approval to implement these new approaches, but the initial approval does not include metagenomic sequencing from host samples.

We note that the updated version of the manuscript contains immunohistochemistry staining of catheters collected from Patients 92 and 132 in Fig 6BE and Figs S4 and S5, which are a major focus of the study. For this approach, bias is introduced through the selection of the primary antibodies to detect and localize the species of interest.

2) I understand that the same species were isolated on more than one occurrence from the same patient. Other studies have shown that bacteria can evolve in the host, especially under antibiotic pressure. Given that the authors already have the bacterial samples in their possession, it would add great value to this manuscript if they showed whether any of the infecting species evolved over time in the presence/absence of antibiotic treatment or whether different strains of the same species were present in patients.

As noted in lines 296-322 of the updated manuscript, we identified several patients who were prescribed antibiotics through several collection periods where we also observed the persistence of one or more bacterial species, including from Patients 82, 92, 97, 100, 104, 115, 123, 127, 130, and 133 (see Sup. Fig. 1B, I, L, O, P, V, DD, HH, KK, MM). To address the Reviewer's comment regarding within-host evolution, we noted that for Patient 104 the opportunistic pathogen *S. marcescens* was present at every collection period despite the fact that the patient was prescribed the β -lactam antibiotic amoxicillin with the β -lactamase inhibitor at 5 out of 6 collection periods. As β -lactam antibiotics are one of the most commonly prescribed classes of antibiotics, we tested whether the repeated exposure of amoxicillin/clavulanic acid would affect the resistance of the isolates to newer generations of β -lactam antibiotics over time. We observed that all of the isolates were resistant to amoxicillin and clavulanic acid and demonstrated resistance to cefazolin and ceftriaxone. Interestingly, we observed that while the first isolate (from collection period 0), was quite sensitive to cefepime and meropenem with MICs at 0.5 and 0.25 $\mu\text{g}/\text{mL}$, respectively, the last isolate (from collection period 6) had MICs of 2 and 1.5 $\mu\text{g}/\text{mL}$, which is approaching the cutoff for clinical levels of resistance.

3) In Fig2C, the authors are showing a summary of the isolation frequency of different species over the different collection time-points. However, it is unclear whether the sample collection interval (eg. Number of days post catheterization) was similar for all recruited patients. Moreover, the authors summarize data from a patient group with different underlying pathologies, age, sex and antibiotic treatment, which makes difficult to draw any conclusion. Did the authors find any associations between sex, age, underlying pathologies, antibiotic treatment and polymicrobial community composition? Please include this information in the manuscript if available. Moreover, I suggest to the authors to show the polymicrobial composition for each patient individually over time, as well as include information such as the onset of antibiotic treatment and the class of antibiotic used.

As the Reviewer requests, we now provide a detailed chart of the polymicrobial community composition at each collection point, which includes information on the duration of the collection period, any time missed between collection periods (due to hospitalization, etc), whether the isolate was collected in the catheter, urine, or at both sites, and the antibiotics that were prescribed to the patient during the specific collection period.

We also limit the updated Figure 2B to only include patients that have 9+ collections.

4) I understand that the authors have stored urine samples collected, which would be a great opportunity to investigate how polymicrobial communities influence the immune response. Did

they perform any urine analysis to quantify host factors, such as cytokines, and if so, can they include any such findings in their study?

This is an excellent suggestion but due to lack of sufficient urine samples available at this point, this will be the subject of future studies. The focus and design of this longitudinal clinical study was to understand the changes of the microbial communities in long-term catheterized patients. We agree with the Reviewer that future clinical studies should be designed to assess the host inflammatory and immune response during infection and their role in polymicrobial community composition. This has now been discussed in the discussion section of the manuscript in lines 380-382.

Minor comments

It is unclear what the counts on the Y axes of figure 3A and B correspond to. Kindly clarify.

The count axes have been clarified in the figure legends as “Number of occurrences of indicated species per patient per collection period detected in the catheter and/or urine samples” in Fig 3A and Fig S2C (formerly figure 3B).

Reviewer #3 (Remarks to the Author):

Microbial co-occurrences on catheters from long-term catheterized patients

General comments: In this manuscript, the authors obtained access to urinary catheters from 55 human patients that were catheterized long-term. Extended follow-up was performed on these patients that often had catheterized replaced on a monthly basis, leading to on average (~ 6.7 samples/patient over the course of the study). Portions of the catheter were inoculated into BHI and then plated on BHI agar for colony identification. Urine samples from the same patient were also grown on BHI agar for colony identification. In total 366 samples were evaluated. ~ 80% of the samples were polymicrobial in nature. The authors did an analysis of co-occurrences in the polymicrobial interactions, and found 19 positive associations and 20 negative associations among the microbial cohorts.

The major take home of this study is that the observations made about positive and negative associations among microbes is that these interactions are very much context dependent, and refute recent studies where media and culture dynamics don't often resemble the in vivo micro-environment. In particular, they focused on *Enterococcus faecalis* as this bacteria was present in 187/366 samples. The next closest species in abundance was *Escherichia coli* in 102/366 samples. Noteworthy was the negative association displayed by *E. faecalis* with other Gram-positive species and the fungal pathogen, *Candida albicans*, and the positive association with predominant Gram-negative species, including *E. coli*, *Enterobacter* sp., *Klebsiella oxytoca* and *Pseudomonas aeruginosa*.

Focusing on the positive association between *E. faecalis* and *E. coli*, the authors performed mono and co-culture analysis in Artificial Urine Media (AUM). In AUM, *E. faecalis* growth was augmented 2 orders of magnitude in the presence of *E. coli*.

Overall, the study provides a clinical snapshot of microbial interactions in the context of urinary tract infection during long-term catheterization. Although no mechanistic insights were provided as to why certain microbial interactions occurred, the study nevertheless provides a roadmap for future studies.

The major critique of the study is why the authors chose culture-based approaches to determine species composition growing on the catheter and not a direct sequencing approach. It would appear that the culture-based approach may exclude lower abundant, but important UTI community members that may contribute to the course of infection.

We agree with the Reviewer that a culture-independent metagenomic approach would have provided a more precise description of the long-term catheterized patients' polymicrobial communities. However, our current study was designed to reflect the practices currently used as the gold standard in clinical microbiology labs, to be reflective of the information that could be gathered and subsequently applied in a hospital setting, where metagenomic sequencing is not routine. Additionally, at the time of the initial study design, patients were not consented for metagenomic sequencing from urine or catheter samples, which would contain host genomic DNA and therefore be subject to new IRB approval. Although the samples were saved in DNA/RNA shield, the purpose of these samples as detailed in the IRB was for the performance of future transcriptomic studies, which are more amenable to host nucleic acid depletion techniques than DNA samples. We have been working to update the study design and the IRB approval to implement these new approaches, but the initial approval does not include metagenomic sequencing from host samples.

Specific comments:

Page 5, lines 10-12: This sentence seems redundant. 80% are polymicrobial with the majority containing 2-3 detected bacterial species. By definition, polymicrobial indicates 2 or more. Please consider restating this sentence.

Our intent is to convey that although the samples were predominately polymicrobial, they only contained 2-3 species per sample as opposed to 4+, which raises intriguing questions about the species dynamics. To better reflect this point, the statement has been changed to:

"We discovered that ~80% of the collected samples are polymicrobial, however most of these samples only contained 2-3 detected bacterial species."

Page 6, lines 6-7. It is a minor point, but when it comes to UTI and catheterization, gender identification seems to take a backseat to biological sex. If the patients are male or female, it seems appropriate to not mention how they identify for a biological variable.

We have removed gender identification from the text, it now reads “We enrolled 55 patients, 20 females and 35 males, at an average age of 64 years”.

Page 7, line 9 as well as other places in the manuscript, the use of enterococcal or staphylococcal as adjectives should not be capitalized. A more appropriate way to deal with this sentence and to be consistent with Proteus and Pseudomonas species would be to state it as Enterococcus, Staphylococcus, Proteus, and Pseudomonas species.

We have updated the genus names and adjectives throughout the manuscript accordingly.

In the methods section under sample preparation, it is mentioned that the remaining portions of the catheter, were split evenly with one half used for immunohistochemistry and the other half stored in DNA shield. No further mention is made of performing immunohistochemistry or genomic DNA extractions from the catheter. Since no downstream analysis was performed on these samples, it seems appropriate to omit this detail.

The updated version of the manuscript includes immunohistochemistry experiments on the fixed catheter samples. We have removed the details regarding the portion saved in DNA/RNA shield.

REVIEWER COMMENTS

Reviewer #1 (Remarks to the Author):

The authors have done a thorough and commendable job addressing all review comments. In particular, inclusion of all longitudinal data and antibiotic exposure for as supplemental images provides a truly impressive overview of species concordance in urine vs catheter specimens that may impact culture practices. My only remaining comment is that Figure 2C in the “response to reviews” shows color coding, while that panel of figure two appears to be the old version.

Reviewer #2 (Remarks to the Author):

I would like to thank the authors for their response to my comments. Please find below my comments to your responses as well as some additional comments.

Previous Major comments

1)Reviewer: The major weakness of the current study is the culture-dependent isolation of the CAUTI causing pathogens. Bacteria were isolated after subculturing or plating the samples in only one media (BHI) under aerobic conditions. The use of only one culture condition might have led to the underestimation of the species number comprising the polymicrobial communities in CAUTI. Moreover, by subculturing the urine samples, information on the relative abundance of each species at different time-points during catheterization is lost. The application of a culture-independent approach, such as metagenomics would have provided a more precise description of the polymicrobial community composition in CAUTI patients, as well as the relative abundance of each species in these communities over time. After reading the methods section of the current manuscript, I understand that the authors have collected catheter samples for immunohistochemistry and DNA isolation. The analysis and inclusion of these samples either with FISH or genome sequencing would add a lot of value to this manuscript because it will reveal the microbial diversity in a culture-independent manner and provide insight into the relative abundance of each species. I recommend that the authors revise their study design accordingly.

Authors: We agree with the Reviewer that a culture-independent metagenomic approach would have provided a more precise description of the long-term catheterized patients' polymicrobial communities. However, our current study was designed to reflect the practices currently used as the gold standard in clinical microbiology labs, to be reflective of the information that could be gathered and subsequently applied in a hospital setting, where metagenomic sequencing is not routine. Additionally, at the time of the initial study design, patients were not consented for metagenomic sequencing from urine or

catheter samples, which would contain host genomic DNA and therefore be subject to new IRB approval. Although the samples were saved in DNA/RNA shield, the purpose of these samples as detailed in the IRB was for the performance of future transcriptomic studies, which are more amenable to host nucleic acid depletion techniques than DNA samples. We have been working to update the study design and the IRB approval to implement these new approaches, but the initial approval does not include metagenomic sequencing from host samples.

Reviewer: It is unfortunate that the authors could not apply a metagenomics approach as it did not allow the study's full novelty potential to be realized. However, it is understandable that they were limited to do so based on their current IRB protocol. I appreciate that the authors discussed the limitation of following a culture-dependent approach in the discussion section.

Authors: We note that the updated version of the manuscript contains immunohistochemistry staining of catheters collected from Patients 92 and 132 in Fig 6BE and Figs S4 and S5, which are a major focus of the study. For this approach, bias is introduced through the selection of the primary antibodies to detect and localize the species of interest.

Reviewer: I thank the authors for adding immunohistochemistry staining in the manuscript. It is very interesting to see the spatial co-localization of *E. coli* and *E. faecalis* on catheters, which are the main focus of this manuscript. These stainings further strengthen the findings of the in vitro co-culturing experiments.

2)Reviewer: I understand that the same species were isolated on more than one occurrence from the same patient. Other studies have shown that bacteria can evolve in the host, especially under antibiotic pressure. Given that the authors already have the bacterial samples in their possession, it would add great value to this manuscript if they showed whether any of the infecting species evolved over time in the presence/absence of antibiotic treatment or whether different strains of the same species were present in patients.

Authors: As noted in lines 296-322 of the updated manuscript, we identified several patients who were prescribed antibiotics through several collection periods where we also observed the persistence of one or more bacterial species, including from Patients 82, 92, 97, 100, 104, 115, 123, 127, 130, and 133 (see Sup. Fig. 1B, I, L, O, P, V, DD, HH, KK, MM). To address the Reviewer's comment regarding within-host evolution, we noted that for Patient 104 the opportunistic pathogen *S. marcescens* was present at every collection period despite the fact that the patient was prescribed the β -lactam antibiotic amoxicillin with the β -lactamase inhibitor at 5 out of 6 collection periods. As β -lactam antibiotics are one of the most commonly prescribed classes of antibiotics, we tested whether the repeated exposure of amoxicillin/clavulanic acid would affect the resistance of the isolates to newer generations of β -lactam antibiotics over time. We observed that all of the isolates were resistant to amoxicillin and clavulanic acid and demonstrated resistance to cefazolin and ceftriaxone. Interestingly, we observed that while the first isolate (from collection period 0), was quite sensitive to cefepime and meropenem with MICs at 0.5 and 0.25 $\mu\text{g}/\text{mL}$, respectively, the last isolate (from collection period 6) had MICs of 2 and 1.5 $\mu\text{g}/\text{mL}$, which is approaching the cutoff for clinical levels of resistance.

Reviewer: I thank the authors for providing this additional data. Perhaps they can consider adding this section in the results instead of the discussion section.

3) Reviewer: In Fig2C, the authors are showing a summary of the isolation frequency of different species over the different collection time-points. However, it is unclear whether the sample collection interval (eg. Number of days post catheterization) was similar for all recruited patients. Moreover, the authors summarize data from a patient group with different underlying pathologies, age, sex and antibiotic treatment, which makes difficult to draw any conclusion. Did the authors find any associations between sex, age, underlying pathologies, antibiotic treatment and polymicrobial community composition? Please include this information in the manuscript if available. Moreover, I suggest to the authors to show the polymicrobial composition for each patient individually over time, as well as include information such as the onset of antibiotic treatment and the class of antibiotic used.

Authors: As the Reviewer requests, we now provide a detailed chart of the polymicrobial community composition at each collection point, which includes information on the duration of the collection period, any time missed between collection periods (due to hospitalization, etc), whether the isolate was collected in the catheter, urine, or at both sites, and the antibiotics that were prescribed to the patient during the specific collection period. We also limit the updated Figure 2B to only include patients that have 9+ collections.

Reviewer: I welcome the addition of supplementary figure 1, which addresses my comment and provides a detailed view of the polymicrobial composition for each patient. This data will be of great interest to people in the field of CAUTI.

I would also like to reiterate my previous question on whether the authors found any associations between sex, age, underlying pathologies, antibiotic treatment and polymicrobial community composition. Did the authors draw any other conclusions from this data besides the different pathogen co-occurrences?

4) Reviewer: I understand that the authors have stored urine samples collected, which would be a great opportunity to investigate how polymicrobial communities influence the immune response. Did they perform any urine analysis to quantify host factors, such as cytokines, and if so, can they include any such findings in their study?

Authors: This is an excellent suggestion but due to lack of sufficient urine samples available at this point, this will be the subject of future studies. The focus and design of this longitudinal clinical study was to understand the changes of the microbial communities in long-term catheterized patients. We agree with the Reviewer that future clinical studies should be designed to assess the host inflammatory and immune response during infection and their role in polymicrobial community composition. This has now been discussed in the discussion section of the manuscript in lines 380-382.

Reviewer: Thank you for your answer. I have no further comments.

Minor comments

Reviewer: It is unclear what the counts on the Y axes of figure 3A and B correspond to. Kindly clarify.

Authors: The count axes have been clarified in the figure legends as “Number of occurrences of indicated species per patient per collection period detected in the catheter and/or urine samples” in Fig 3A and Fig S2C (formerly figure 3B).

Reviewer: Thank you for clarifying.

New major comments

Comment 1: In the original manuscript, the authors evaluated the different bacterial co-occurrences, which revealed that *E. faecalis* is positively co-associated with *E. coli*. This finding justified the second part of the manuscript in which the authors explore the interaction between these 2 species in vitro. Reviewer 1 pointed out an error in the statistical analysis of co-occurrences during the first round of revision. To correct this error, the authors re-analyzed their data and replaced the original figure 4 with a new one. If I understand correctly, the new co-occurrence analysis did not identify a positive co-occurrence between *E. coli* and *E. faecalis* and the decision to further investigate the interaction between these 2 species is justified by observing their co-occurrence in patient 132 (lines 224-227). Selecting to study *E. coli* and *E. faecalis* interaction based on the co-occurrence in patient 132 seems arbitrary when there are so many other statistically significant negative or positive co-occurrences identified in this study.

I encourage the authors to reflect on the structure, and take-home message of the current version of the manuscript. Currently the manuscript begins as a descriptive study of CAUTI samples. The authors present an impressive amount of clinical data that will undoubtedly be of great interest to researchers in the field. They also draw interesting conclusions on pathogen co-occurrences. However, none of the experiments in the rest of the manuscript explore these co-occurrences.

My suggestion to make this manuscript more balanced is that the authors consider exploring in vitro some of the significant co-occurrences identified in figure 4, besides the *E. faecalis*-*E. coli* interaction.

Comment 2: *P. aeruginosa* is present in the urine in almost every sample collected from patient 132. This suggests that *P. aeruginosa* might also influence the occurrence observed in that patient. It would be very interesting to include co-culture experiments of *E. faecalis* and *E. coli* with the *P. aeruginosa* strain isolated from that patient.

New Minor comments:

1. The X-axis is missing for plot A in supplementary figure 1 in my version. Kindly double-check this plot.
2. Figure 5b: Is the red circle supposed to include only some of the data points of patient 132? Shouldn't it include all data points. It is not clear what the red circle is supposed to highlight.

Latest Reviewer comments are in black

Previous Reviewer and Author comments are in blue

Author responses are in red

REVIEWER COMMENTS

Reviewer #1 (Remarks to the Author):

The authors have done a thorough and commendable job addressing all review comments. In particular, inclusion of all longitudinal data and antibiotic exposure for as supplemental images provides a truly impressive overview of species concordance in urine vs catheter specimens that may impact culture practices. My only remaining comment is that Figure 2C in the “response to reviews” shows color coding, while that panel of figure two appears to be the old version.

We thank the reviewer for their kind comments on our revision. We have ensured that a color-coded version of Figure 2C has been submitted.

Reviewer #2 (Remarks to the Author):

I would like to thank the authors for their response to my comments. Please find below my comments to your responses as well as some additional comments.

Previous Major comments

1) **Reviewer:** The major weakness of the current study is the culture-dependent isolation of the CAUTI causing pathogens. Bacteria were isolated after subculturing or plating the samples in only one media (BHI) under aerobic conditions. The use of only one culture condition might have led to the underestimation of the species number comprising the polymicrobial communities in CAUTI. Moreover, by subculturing the urine samples, information on the relative abundance of each species at different time-points during catheterization is lost. The application of a culture-independent approach, such as metagenomics would have provided a more precise description of the polymicrobial community composition in CAUTI patients, as well as the relative abundance of each species in these communities over time. After reading the methods section of the current manuscript, I understand that the authors have collected catheter samples for immunohistochemistry and DNA isolation. The analysis and inclusion of these samples either with FISH or genome sequencing would add a lot of value to this manuscript because it will reveal the microbial diversity in a culture-independent manner and provide insight into the relative abundance of each species. I recommend that the authors revise their study design accordingly.

Authors: We agree with the Reviewer that a culture-independent metagenomic approach would have provided a more precise description of the long-term catheterized patients' polymicrobial communities. However, our current study was designed to reflect the practices currently used as the gold standard in clinical microbiology labs, to be reflective of the information that could be

gathered and subsequently applied in a hospital setting, where metagenomic sequencing is not routine. Additionally, at the time of the initial study design, patients were not consented for metagenomic sequencing from urine or catheter samples, which would contain host genomic DNA and therefore be subject to new IRB approval. Although the samples were saved in DNA/RNA shield, the purpose of these samples as detailed in the IRB was for the performance of future transcriptomic studies, which are more amenable to host nucleic acid depletion techniques than DNA samples. We have been working to update the study design and the IRB approval to implement these new approaches, but the initial approval does not include metagenomic sequencing from host samples.

Reviewer: It is unfortunate that the authors could not apply a metagenomics approach as it did not allow the study's full novelty potential to be realized. However, it is understandable that they were limited to do so based on their current IRB protocol. I appreciate that the authors discussed the limitation of following a culture-dependent approach in the discussion section.

We thank the reviewer for their understanding.

2) **Authors:** We note that the updated version of the manuscript contains immunohistochemistry staining of catheters collected from Patients 92 and 132 in Fig 6BE and Figs S4 and S5, which are a major focus of the study. For this approach, bias is introduced through the selection of the primary antibodies to detect and localize the species of interest.

Reviewer: I thank the authors for adding immunohistochemistry staining in the manuscript. It is very interesting to see the spatial co-localization of *E. coli* and *E. faecalis* on catheters, which are the main focus of this manuscript. These stainings further strengthen the findings of the in vitro co-culturing experiments.

We agree with the reviewer that the immunohistochemistry strengthened the manuscript and thank them for their suggestion.

2) **Reviewer:** I understand that the same species were isolated on more than one occurrence from the same patient. Other studies have shown that bacteria can evolve in the host, especially under antibiotic pressure. Given that the authors already have the bacterial samples in their possession, it would add great value to this manuscript if they showed whether any of the infecting species evolved over time in the presence/absence of antibiotic treatment or whether different strains of the same species were present in patients.

Authors: As noted in lines 296-322 of the updated manuscript, we identified several patients who were prescribed antibiotics through several collection periods where we also observed the persistence of one or more bacterial species, including from Patients 82, 92, 97, 100, 104, 115, 123, 127, 130, and 133 (see Sup. Fig. 1B, I, L, O, P, V, DD, HH, KK, MM). To address the Reviewer's comment regarding within-host evolution, we noted that for Patient 104 the opportunistic pathogen *S. marcescens* was present at every collection period despite the fact that the patient was prescribed the β -lactam antibiotic amoxicillin with the β -lactamase inhibitor at 5 out of 6 collection periods. As β -lactam antibiotics are one of the most commonly

prescribed classes of antibiotics, we tested whether the repeated exposure of amoxicillin/clavulanic acid would affect the resistance of the isolates to newer generations of β -lactam antibiotics over time. We observed that all of the isolates were resistant to amoxicillin and clavulanic acid and demonstrated resistance to cefazolin and ceftriaxone. Interestingly, we observed that while the first isolate (from collection period 0), was quite sensitive to cefepime and meropenem with MICs at 0.5 and 0.25 $\mu\text{g}/\text{mL}$, respectively, the last isolate (from collection period 6) had MICs of 2 and 1.5 $\mu\text{g}/\text{mL}$, which is approaching the cutoff for clinical levels of resistance.

Reviewer: I thank the authors for providing this additional data. Perhaps they can consider adding this section in the results instead of the discussion section.

We chose to add the results of Table 1 in the discussion because although it provides an interesting example of apparent within-host evolution upon chronic exposure to antibiotics, which justifies future study of a larger patient cohort, the data are not sufficient to be conclusive with $n=1$. Thus, future controlled studies are necessary to make any definitive conclusions about the evolution of uropathogens in long-term catheterized patients with chronic antibiotic use.

4) I welcome the addition of supplementary figure 1, which addresses my comment and provides a detailed view of the polymicrobial composition for each patient. This data will be of great interest to people in the field of CAUTI.

I would also like to reiterate my previous question on whether the authors found any associations between sex, age, underlying pathologies, antibiotic treatment and polymicrobial community composition. Did the authors draw any other conclusions from this data besides the different pathogen co-occurrences?

Per the reviewer's request, we examined which associations could be tested robustly by our dataset. We examined the relationship between age and antibiotic use and found no significant relationship between them (p -value= 0.1494, Fisher's exact test). While we agree the association between sex, age, underlying pathologies and antibiotic treatment with polymicrobial community composition would be interesting, our patient demographics and the size/diversity of our dataset severely limit the validity of these analyses. These analyses would need to be performed at the patient level as opposed to individual collection periods due to the previously discussed issues with independence of samples. When we look at the patient-level, we only have one patient (Patient 87, with a total of one collection period) that is monomicrobial, 51% of patients have at least one monomicrobial collection period among the polymicrobial collection periods, and the rest of the patients are polymicrobial at all collection points. Thus, it is difficult to determine how to treat the polymicrobial community composition for each patient given the within-patient diversity across collection periods. Further, our patient cohort consists of older patients (average age 63.4 ± 15.5 years) with many comorbidities, which would require a larger number of enrolled patients in order to perform statistically significant analyses.

4) Reviewer: I understand that the authors have stored urine samples collected, which would be a great opportunity to investigate how polymicrobial communities influence the immune

response. Did they perform any urine analysis to quantify host factors, such as cytokines, and if so, can they include any such findings in their study?

Authors: This is an excellent suggestion but due to lack of sufficient urine samples available at this point, this will be the subject of future studies. The focus and design of this longitudinal clinical study was to understand the changes of the microbial communities in long-term catheterized patients. We agree with the Reviewer that future clinical studies should be designed to assess the host inflammatory and immune response during infection and their role in polymicrobial community composition. This has now been discussed in the discussion section of the manuscript in lines 380-382.

Reviewer: Thank you for your answer. I have no further comments.

Minor comments

Reviewer: It is unclear what the counts on the Y axes of figure 3A and B correspond to. Kindly clarify.

Authors: The count axes have been clarified in the figure legends as “Number of occurrences of indicated species per patient per collection period detected in the catheter and/or urine samples” in Fig 3A and Fig S2C (formerly figure 3B).

Reviewer: Thank you for clarifying.

New major comments

Comment 1: In the original manuscript, the authors evaluated the different bacterial co-occurrences, which revealed that *E. faecalis* is positively co-associated with *E. coli*. This finding justified the second part of the manuscript in which the authors explore the interaction between these 2 species in vitro. Reviewer 1 pointed out an error in the statistical analysis of co-occurrences during the first round of revision. To correct this error, the authors re-analyzed their data and replaced the original figure 4 with a new one. If I understand correctly, the new co-occurrence analysis did not identify a positive co-occurrence between *E. coli* and *E. faecalis* and the decision to further investigate the interaction between these 2 species is justified by observing their co-occurrence in patient 132 (lines 224-227). Selecting to study *E. coli* and *E. faecalis* interaction based on the co-occurrence in patient 132 seems arbitrary when there are so many other statistically significant negative or positive co-occurrences identified in this study.

We appreciate the thoughtful comments and the opportunity to clarify our design of this test. We chose to study the interaction between *E. coli* and *E. faecalis* because they were the most commonly occurring species identified among our whole patient cohort (**Figure 3A**). In parallel, our co-occurrence analysis looked at positive interactions at each independent collection period, which were identified by the number of observed co-occurrences being greater than expected, in which we reasoned that a positive interaction between the most abundant species might be missed due to high expected co-occurrence due to high abundance. Additionally, longitudinal co-colonization is also taken into consideration, and *E. coli* and *E. faecalis* are found to co-colonize the urinary tract of Patients 132 and 92 in almost all collection periods. Isolate abundance, co-occurrence, and longitudinal co-colonization were considered parallelly to select

the strains for interaction studies. In the scheme of characterizing and treating polymicrobial infections, we reasoned it would be of interest to examine the species pairs that occur most commonly and longitudinally in the long-term patient cohort, which is why we proceeded to investigate the interaction between *E. coli* and *E. faecalis*.

We thank the reviewer again for the thoughtful suggestion and have now provided more interaction results between other bacterial species. Besides testing the *E. faecalis-E. coli* interaction, we now have conducted more tests on *E. faecalis-K. pneumoniae* and *E. faecalis-P. aeruginosa* and have provided the results as the new Supplementary Figure 8 and Supplementary Figure 9, as well as new Table S15 and Table S16. More details about these new interaction tests are also described in the response section below.

I encourage the authors to reflect on the structure, and take-home message of the current version of the manuscript. Currently the manuscript begins as a descriptive study of CAUTI samples. The authors present an impressive amount of clinical data that will undoubtedly be of great interest to researchers in the field. They also draw interesting conclusions on pathogen co-occurrences. However, none of the experiments in the rest of the manuscript explore these co-occurrences.

My suggestion to make this manuscript more balanced is that the authors consider exploring *in vitro* some of the significant co-occurrences identified in figure 4, besides the *E. faecalis-E.coli* interaction.

We thank the reviewer for the thoughtful suggestion. We now provide more bacterial interaction results by characterizing the interaction between *E. faecalis* and *K. pneumoniae* strains. *E. faecalis* and *K. pneumoniae* is identified with positive co-occurrence in our co-occurrence analysis at collection period 10 (**Figure 4K**). We choose four pairs of these two species at collection periods 10 and 11 from Patients 92 and 100 for *in vitro* bacterial interaction test. The results are provided as the new Supplementary Figure 8 and Table S15. We have also edited the associated Results section in Line 268 on Page 12 and Materials and Methods section in Line 488 on Page 22.

Comment 2: *P. aeruginosa* is present in the urine in almost every sample collected from patient 132. This suggests that *P. aeruginosa* might also influence the occurrence observed in that patient. It would be very interesting to include co-culture experiments of *E. faecalis* and *E. coli* with the *P. aeruginosa* strain isolated from that patient.

We thank the reviewer for the thoughtful suggestion. We now provide more bacterial interaction results by characterizing the interaction between *E. faecalis* and *P. aeruginosa* strains. We choose these two species at collection periods 00 and 01 from Patient 132 for *in vitro* bacterial interaction test, and provide the results as the new Supplementary Figure 9 and Table S16. We have also edited the associated Results section in Line 268 on Page 12 and Materials and Methods section in Line 488 on Page 22.

New Minor comments:

1. The X-axis is missing for plot A in supplementary figure 1 in my version. Kindly double-check this plot.

We thank the reviewer for bringing this to our attention. It has been corrected in the revised version of the manuscript.

2. Figure 5b: Is the red circle supposed to include only some of the data points of patient 132? Shouldn't it include all data points. It is not clear what the red circle is supposed to highlight.

The red circle is meant to include all data points for Patient 132, although some of the points overlap because the community composition was identical (see Figure 6A). This point has been clarified in the figure legend text with the addition of "Some individual collection points overlap on the plot where community composition was very similar/identical."

REVIEWERS' COMMENTS

Reviewer #2 (Remarks to the Author):

I would like to thank the authors for addressing my new major and minor comments. I have no further comments.